# Stretchable and negative-Poisson-ratio porous metamaterials

Xiaoyu Zhang [1], Qi Sun[1], Xing Liang [1], Puzhong Gu[1], Zhenyu Hu[1], Xiao Yang[1], Muxiang Liu[1], Zejun Sun[1], Jia Huang [1], Guangming Wu[2] & Guoqing Zu [1] ✉

Highly stretchable porous materials are promising for flexible electronics but their fabrication is a great challenge. Herein, several kinds of highly stretchable conductive porous elastomers with low or negative Poisson's ratios are achieved by uniaxial, biaxial, and triaxial hot-pressing strategies. The reduced graphene oxide/polymer nanocomposite elastomers with folded porous structures obtained by uniaxial hot pressing exhibit high stretchability up to 1200% strain. Furthermore, the meta-elastomers with reentrant porous structures combining high biaxial (or triaxial) stretchability and negative Poisson's ratios are achieved by biaxial (or triaxial) hot pressing. The resulting elastomer-based wearable strain sensors exhibit an ultrawide response range (0-1200%). The materials can be applied for smart thermal management and electromagnetic interference shielding, which are achieved by regulating the porous microstructures via stretching. This work provides a versatile strategy to highly stretchable and negative-Poisson-ratio porous materials with promising features for various applications such as flexible electronics, thermal management, electromagnetic shielding, and energy storage.

Porous materials such as aerogels, foams, and sponges possess many unique mechanical, thermal, electrical, and chemical properties, and show promising applications in thermal insulation, adsorption, sensors, catalysis, energy storage, etc.[1–3]. Various flexible porous materials including compressible, bendable, and stretchable porous materials have been developed by optimizing their porous microstructures[4–6]. Among them, stretchable aerogels and foams are especially attractive because of their potential applications in flexible strain/pressure sensors, stretchable conductors, flexible batteries, flexible supercapacitors, stretchable electromagnetic interference (EMI) shielding materials, stretchable thermal management materials, etc.[6–8]. While the compressible and bendable aerogels and foams have been extensively studied and reported, there are much fewer reports on stretchable aerogels and foams. Many specially designed structures such as cellular and layered structures can endow porous materials with high compressibility and elasticity[4,5,7–12]. However, the highly porous structures of aerogels and foams usually tend to be broken when they are stretched. It is a great challenge to achieve highly stretchable aerogels and foams.

Some chemically or physically crosslinked polymer aerogels and foams are stretchable benefiting from the excellent deformability and recoverability of the flexible polymer networks. The reported poly(-isocyanurate-urethane) aerogels can be stretched 120% without fracture[13]. The stretchable polymer aerogels based on poly(caprolactone) exhibit elongations at break of -80–275%[14]. The stretchable (25% elongation at break) polyimide aerogel fibers have been developed by a sol-gel confined transition method for thermal insulation[15]. The regenerated stretchable all-cellulose sponge-aerogel fibers with a graded aligned nanostructure show a maximum tensile strain of 20–50%[16]. The cellulose nanofibers/polyurethane (PU) aerogels exhibit an elongation at break of 10–25%[17]. The chemically crosslinked poly(3,4-ethylenedioxythiophene):polystyrene sulfonate (PEDOT:PSS) networks can endow the PEDOT:PSS foams with good stretchability with an elongation at break of 10–100% and a reversible tensile strain of 40–60%[18].

[1]Interdisciplinary Materials Research Center, Department of Polymeric Materials, School of Materials Science and Engineering, Tongji University, Shanghai 201804, PR China. [2]Shanghai Key Laboratory of Special Artificial Microstructure Materials and Technology, School of Physics Science and Engineering, Tongji University, Shanghai 200092, PR China. ✉e-mail: guoqingzu@tongji.edu.cn

The combination of flexible polymers and inorganic building blocks such as graphene and MXene can afford stretchable organic-inorganic hybrid aerogels and foams. The reported MXene/polyimide aerogels exhibit an elongation at break of 25–35% and a reversible tensile strain of 20%[19]. The "layer-strut" skeletons of reduced graphene oxide (rGO)/polyimide facilitate load transfer between rGO sheets and polyimide during deformation, allowing the rGO/polyimide foams to exhibit an elongation at break of 16%[20]. The aerogel fibers based on aramid nanofiber/carbon nanotube (CNT) can withstand a 20–26% tensile strain[21]. Besides, we have reported stretchable (~100% elongation at break) rGO/PU aerogels with a gradient porous structure for high-performance pressure-sensitive wearable electronics[22].

Special porous microstructures such as lamellar and honeycomb-like structures may effectively reduce stress concentration and contribute to stress transfer, which may make the resultant porous materials stretchable. Porous carbon materials with a long-range lamellar multi-arch structure obtained by bidirectional freezing can achieve a maximum tensile strain of 80%[23]. Graphene aerogels with a hyperboloid structure by hydroplastic foaming exhibit a break elongation of ~20%[24]. Graphene aerogels with a highly crimped and cross-linked network can achieve a reversible elongation of 400%[25].

The porous materials with fibrous structures may be stretchable because of the flexible fibers that compose the networks. Nanofibrous polymer aerogels with a hyperconnective network of aramid nanofiber composites are stretchable (20–30% elongation at break) and can be used for thermal insulation, wearable electronics, and filtration[26]. The silica nanofiber aerogels prepared by combining electrospinning and freeze drying possess good stretchability (~220% elongation at break) and show potential applications in thermal insulation and radiative cooling[27]. The ceramic aerogels composed of curly SiC-SiOx nanofibers show a reversible elongation of 20%[7]. The interwoven crimped nanofibrous structure endows the ceramic nanofibrous aerogels with good stretchability up to 100% tensile strain (40% reversible tensile strain)[28]. Stretchable (withstand a 40% tensile strain) hypocrystalline zircon nanofibrous aerogels with a near-zero Poisson's ratio have been achieved by constructing a zig-zag architecture[29]. The $Si_3N_4$ nanofiber sponge with an interlocked nanostructure show a break elongation of ~80% and a reversible elongation of 20%[30].

Some special macroscopic structures such as the structures of lattice, serpentine, spring, and wrinkle show high deformability and elasticity, which can make the aerogels and foams with these kinds of structures stretchable. The stretchable graphene/CNT aerogel lattices prepared by ink-printing possess a reversible 200% elongation and can be used as strain sensors[6]. The shape memory polymer/graphene aerogel lattices can withstand a 100% tensile strain[31]. While the graphene aerogels with serpentine and spiral structures show high stretchability up to 1200% and 5400%, respectively, the same graphene aerogels without the special macroscopic structures only exhibit an elongation at break of 6%[8]. The stretchability of the reported aerogels is far from satisfactory. In addition, the elasticity and durability of the stretchable porous materials need to be further improved.

Metamaterials with a negative Poisson's ratio contract laterally when compressed and expand laterally when stretched, which can exhibit excellent deformability and toughness[29,32–34]. They have attracted a lot of attention for their unusual mechanical properties and promising applications in shock-absorbing materials, air filters, fasteners, etc. Aerogels and foams with negative Poisson's ratios during compression have been reported[4,8,29,32–36]. The elastic ceramic aerogels with a hyperbolic architecture show negative Poisson's ratios during compression[4]. Compressible polyimide aerogels with negative Poisson's ratios can be achieved by directional and tridirectional freezing strategies[35,36]. Graphene aerogels with Poisson's ratios in the range of −0.95–1.64 during compression can be obtained by constructing meta-structures via laser engraving[8]. However, highly stretchable porous materials with negative Poisson's ratios during stretching are rarely reported.

Herein, we report highly stretchable porous rGO/polymer nanocomposite elastomers with low or negative Poisson's ratios achieved by uniaxial, biaxial, and triaxial hot-pressing strategies. Highly compressible aerogels with positive Poisson's ratios can be converted into highly stretchable porous meta-elastomers with zero or negative Poisson's ratios via these hot-pressing strategies. The uniaxially hot-pressed porous elastomers with compressed and folded porous structures exhibit high stretchability with an elongation at break of 1250% and a reversible elongation larger than 800%. Besides, the porous meta-elastomers with reentrant porous structures obtained by biaxial (or triaxial) hot pressing show high biaxial (or triaxial) stretchability and a negative Poisson's ratio. We demonstrate that the resulting porous elastomers can be applied for ultrabroad-range-response strain (0–1200%) and pressure (0–9.5 MPa) sensors. In addition, they can be applied for smart thermal management and EMI shielding, which are achieved by regulating the porous microstructures simply via stretching. This work opens a way to highly stretchable and negative-Poisson-ratio porous materials with application possibilities in flexible electronics, thermal management, EMI shielding, energy storage, etc.

## Results
### Preparation
The foldable Chinese lantern can be both highly compressed and stretched because of its foldable accordion- or honeycomb-like structure (Fig. 1a). Inspired by the foldable lantern, highly stretchable porous rGO/polymer nanocomposite elastomers have been developed via uniaxial, biaxial, and triaxial hot-pressing strategies (Fig. 1b–d). Graphene oxide (GO) was used as the raw material of rGO, while PU foam (PUF), PU (dispersed in water), polyvinyl alcohol (PVA), or melamine foam (MF) was used as the toughening polymer (Supplementary Fig. 1). Ethanediamine or (3-aminopropyl)triethoxysilane was used as the crosslinker and reductant of GO. Polypyrrole (PPy) was introduced in the aerogels via in situ oxidation polymerization of pyrrole to further enhance their electrical conductivities. The pristine aerogels were prepared via either freeze drying or ambient pressure drying (APD). The starting compositions of typical rGO/polymer aerogels are listed in Supplementary Tables 1–3. The reaction and microstructure variation during preparation are schematically presented in Fig. 1b and Supplementary Fig. 2.

Hot pressing of the rGO/polymer aerogels was readily performed using a home-made apparatus (Fig. 1c and Supplementary Figs. 3–8). In the case of uniaxial hot pressing, a monolithic rGO/polymer aerogel was first sandwiched with two pieces of glass and then compressed with 66.7–87.5% strain in $x$ direction using two other pieces of glass, followed by fixing and heat treatment at 120 or 140 °C (Supplementary Figs. 3 and 6). In the case of biaxial hot pressing, a monolithic aerogel was first compressed 50% in $x$ direction and then compressed 50% in $y$ direction, followed by fixing and heat treatment (Supplementary Figs. 4 and 7). In the case of triaxial hot pressing, a monolithic aerogel was first compressed 50% in $z$ direction and then compressed 50% in both $x$ and $y$ directions, followed by fixing and heat treatment (Supplementary Figs. 5 and 8). After hot pressing, the aerogel-derived porous materials maintained the compressed shapes without springing back when they were released.

In our previous work[22], uniaxial hot pressing is used to enhance the modulus of the high-modulus layer of the gradient rGO/PUF composite aerogels instead of their stretchability. In this work, uniaxial, biaxial, and triaxial hot-pressing strategies are used to enhance stretchability and achieve near-zero and negative Poisson's ratios of the porous materials. Besides, the sensors in this work are stretched in the opposite direction of hot pressing, while the stretching direction of

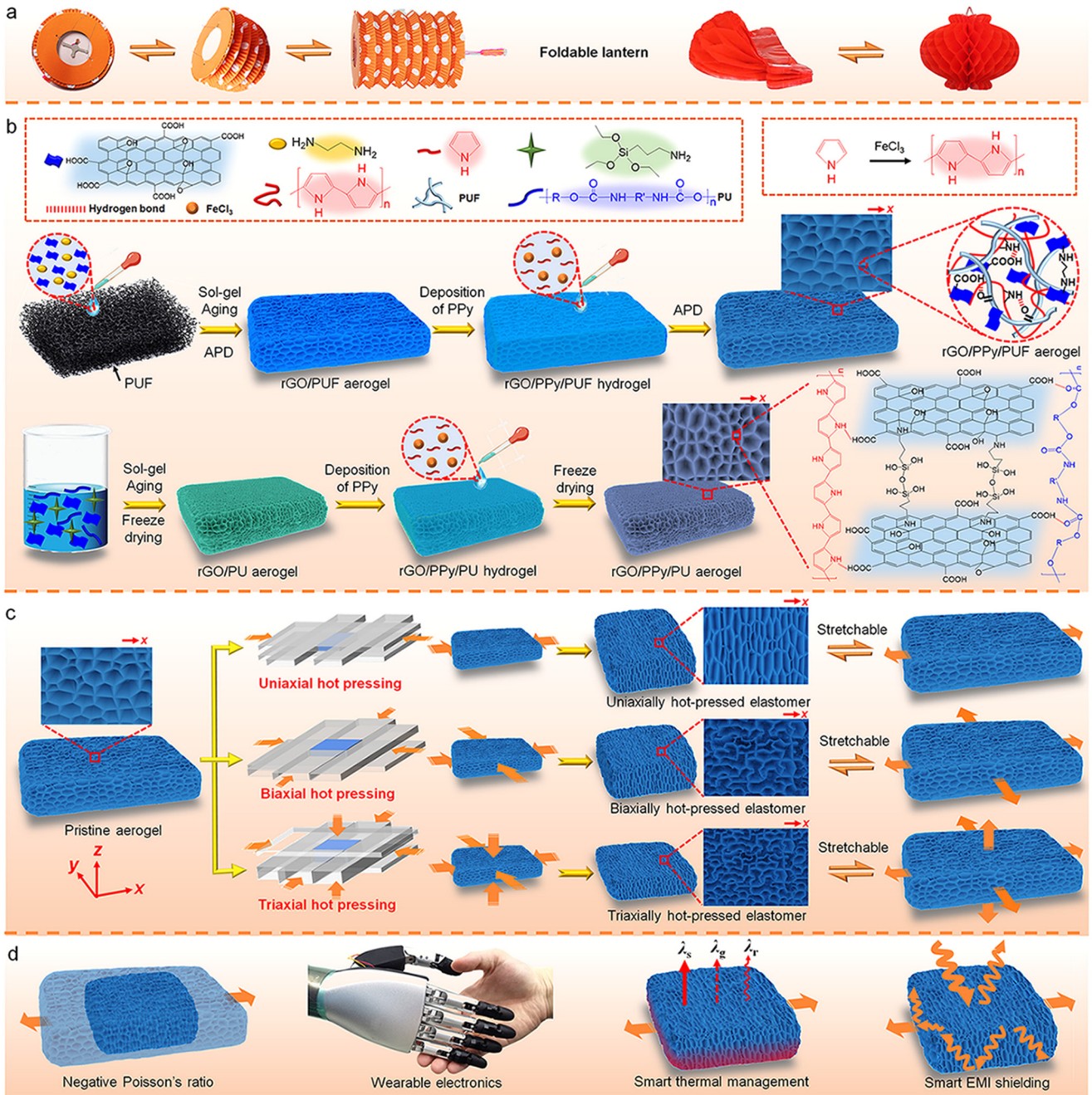

**Fig. 1 | Preparation and applications of the stretchable porous rGO/polymer nanocomposite elastomers. a** Foldable lantern-inspired stretchable structures. **b** Preparation of the rGO/polymer aerogels based on rGO/PPy/PUF and rGO/PPy/ PU. **c** Preparation of the stretchable porous rGO/polymer elastomers via uniaxial, biaxial, and triaxial hot-pressing strategies. **d** Schematic of applications of the stretchable porous elastomers.

the sensors in our previous work is perpendicular to the direction of hot pressing.

## Structural characterization

The appearances of typical rGO/polymer porous materials before and after hot pressing are shown in Fig. 2a–h and Supplementary Figs. 6–8. The pristine rGO/polymer aerogels obtained by introducing different toughening polymers (PUF, PU, PVA, and MF) exhibit highly porous structures with different morphologies (Fig. 2i–t and Supplementary Figs. 9–11). The interconnected rGO nanosheets are observed in rGO/ PPy/PUF1, rGO/PPy/PUF2, and rGO/PVA/MF (Supplementary Figs. 9 and 10). Besides, there are many irregular spherical particles on the skeletons of rGO/PPy/PUF1, rGO/PPy/PUF2, and rGO/PPy/PU, which are

well preserved after hot pressing (Fig. 2i–p and Supplementary Fig. 10). These particles are supposed to be PPy polymers, which are deposited by in situ oxidation polymerization of pyrrole.

The uniaxially hot-pressed porous elastomers (rGO/PPy/PUF1-UHP, rGO/PPy/PUF2-UHP, rGO/PPy/PU-UHP, and rGO/PVA/MF-UHP) exhibit compressed and folded porous structures in the $xy$ and $xz$ planes with smaller pores compared with those of pristine aerogels without hot pressing (Fig. 2j, n, q and Supplementary Figs. 9–11). The biaxially hot-pressed porous elastomers (rGO/PPy/PUF1-BHP and rGO/ PPy/PU-BHP) present reentrant porous structures in the $xy$ plane and compressed and folded porous structures in the $xz$ and $yz$ planes, which are quite different from those of pristine aerogels without hot pressing (Fig. 2k, o, r and Supplementary Fig. 11). The triaxially hot-

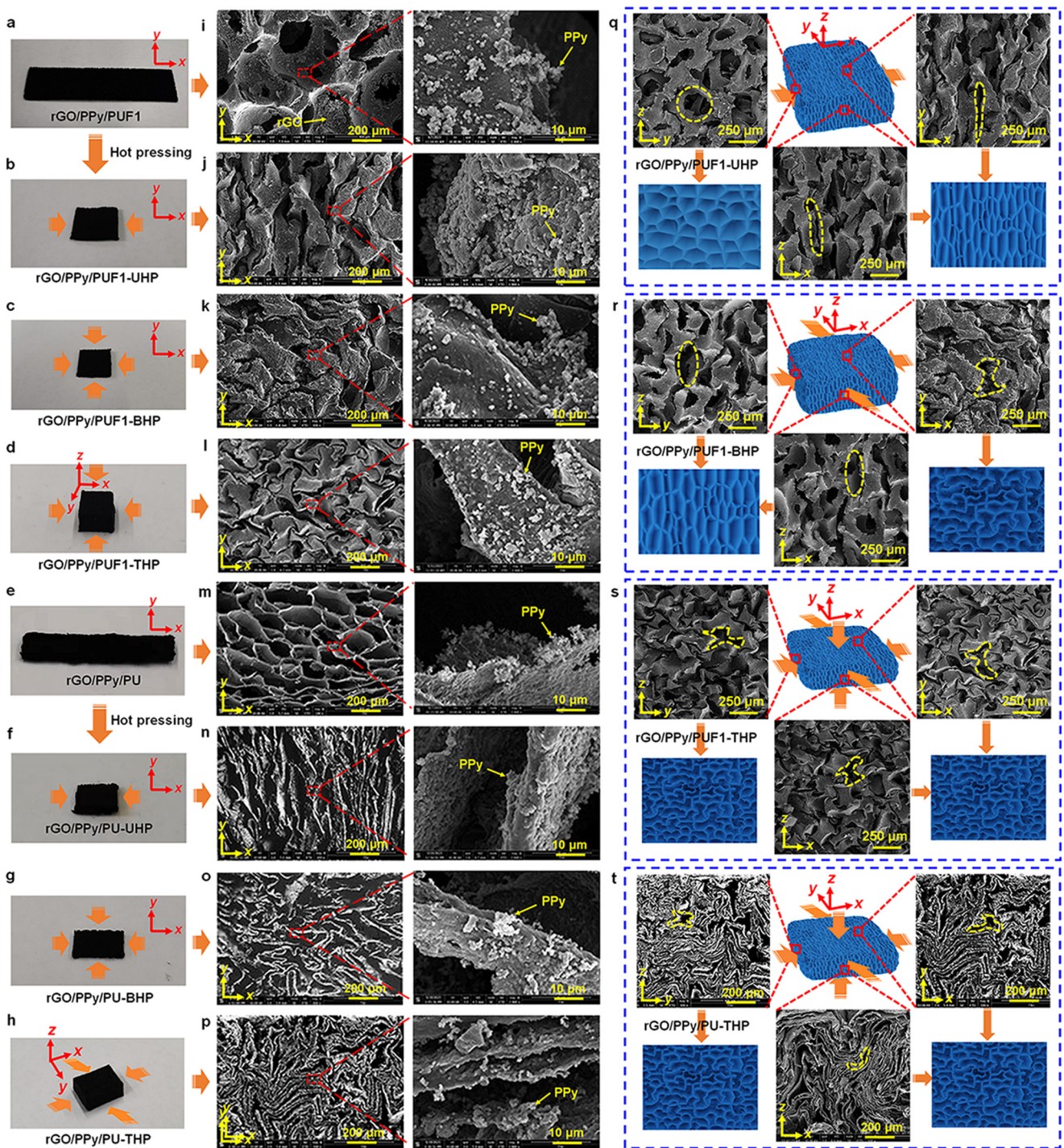

**Fig. 2 | Photographs and morphologies of the porous rGO/polymer nanocomposite elastomers before and after uniaxial, biaxial, and triaxial hot pressing. a–h** Photographs of typical porous elastomers. **i–p** Scanning electron microscope (SEM) images of typical porous elastomers. SEM images of (**q**) uniaxially, (**r**) biaxially, and (**s, t**) triaxially hot-pressed porous elastomers in the *xy*, *xz*, and *yz* planes.

pressed porous elastomers (rGO/PPy/PUF1-THP and rGO/PPy/PU-THP) show the similar compressed and reentrant porous structures in the three planes (*xy*, *xz*, and *yz* planes) (Fig. 2l, p, s, t). Although the bulk density increases and pore size and porosity decrease after hot pressing, the resultant materials still possess porous structures (Supplementary Figs. 12, 13 and Note 1).

Softening and plastic deformations of the polymers (PUF, PU, PVA, or MF) in the skeletons of the rGO/polymer nanocomposites will occur when the porous materials were compressed and heated at 120–140 °C. The polymer chains tend to be deformed and move at high temperatures upon compression, and the shape of the polymer chains after deformation can be fixed when cooling down to room temperature after hot pressing. This will facilitate the shape fixing of the compressed rGO/polymer skeletons after hot pressing. The dynamic thermomechanical analysis (DMA) confirms that rGO/PPy/PUF1 and rGO/PPy/PU undergo a softening process during hot pressing (Supplementary Fig. 14). Besides, as shown in the optical microscope and SEM images (Fig. 2 and Supplementary Figs. 9–11), new contacts between the neighboring pore walls of the elastomers can be produced after hot pressing, which will result in further physical

crosslinking (such as hydrogen bonds) at these contact points. This may also contribute to the shape fixing of the elastomers after hot pressing. Since the skeletons of the rGO/PPy/PUF and rGO/PPy/PU composite materials are mainly composed of PU, their shape fixing processes are supposed to be dominated by the PU matrix. Interestingly, the shape fixing process of these materials are reversible (Supplementary Fig. 15).

To investigate the microstructure variation of the materials during hot pressing, the microstructures of the rGO/polymer composites after uniaxial compression with 75% strain at room temperature (rGO/PPy/PUF1-UP) and the rGO/polymer composites after uniaxial hot pressing with 75% strain at 140 °C (rGO/PPy/PUF1-UHP) were observed by an optical microscope (Supplementary Fig. 16). It is found that there is no obvious difference in microstructures between rGO/PPy/PUF1-UP and rGO/PPy/PUF1-UHP, indicating that the strain release caused by softening process during hot pressing is not obvious and can be negligible.

The chemical structures of the porous rGO/polymer elastomers were investigated by X-ray diffraction (XRD) patterns, Raman spectra, Fourier transform infrared (FTIR) spectra, and X-ray photoelectron spectra (XPS) (Supplementary Figs. 17–20). These investigations confirm the successful incorporation of rGO and PPy in the rGO/polymer elastomers (Supplementary Note 2)[20,22,37,38]. XPS analysis indicates that GO nanosheets can be crosslinked by APTES via a C-N coupling reaction between epoxy groups of GO and amino groups of APTES combined with the hydrolytic polycondensation of alkoxy groups of APTES (Supplementary Figs. 18–20, Table 4 and Note 3). Besides, the interfaces between the polymer and rGO as well as the morphology and crystal size of rGO were investigated (Supplementary Figs. 21, 22 and Supplementary Note 4). It is found that the polymers and rGO are well combined in the rGO/polymer elastomers, which is crucial for achieving excellent mechanical properties.

## Mechanical properties

Benefiting from their special porous structures, the hot-pressed rGO/polymer elastomers exhibit high stretchability and high elasticity (Fig. 3 and Supplementary Figs. 23, 24). The uniaxially hot-pressed elastomers rGO/PPy/PUF1-UHP and rGO/PPy/PUF2-UHP exhibit high stretchability with elongations at break of 810% and 1250%, respectively, and reversible elongations larger than 700% and 800%, respectively (Fig. 3a, d, e and Supplementary Fig. 23 and Movies 1–4). After stretching-releasing with 500% strain for 1000 cycles, rGO/PPy/PUF1-UHP nearly recovered its original shape, indicating the high elasticity and fatigue resistance of the elastomers (Fig. 3f). rGO/PPy/PU-UHP and rGO/PVA/MF-UHP also exhibit high stretchability in $x$ direction with elongations at break of 470% and 112%, respectively, and reversible elongations of 400% and 100%, respectively (Fig. 3d, g and Supplementary Fig. 24). The stretchability of the hot-pressed elastomers is significantly higher than that of the pristine aerogels without hot pressing (Supplementary Figs. 25 and 26).

In addition, the biaxially hot-pressed rGO/polymer elastomers possess high biaxial stretchability. For example, rGO/PPy/PUF1-BHP exhibits elongations at break of 335–340% and reversible elongations of 300% in both $x$ and $y$ directions (Fig. 3b, d, h and Supplementary Movies 5, 6). Furthermore, the triaxially hot-pressed rGO/polymer elastomers present high triaxial stretchability. The elongations at break of rGO/PPy/PUF1-THP are in the range of 310–340% in $x$, $y$, and $z$ directions (Fig. 3d and Supplementary Movie 7).

In order to further evaluate the stretchability of the hot-pressed elastomers, the theoretical values of the elongations at break of typical elastomers were calculated. The calculation method and results are illustrated in Supplementary Figs. 27–29. The calculated theoretical values of the elongations at break of rGO/PPy/PUF1-UHP, rGO/PPy/PUF2-UHP, and rGO/PPy/PUF1-BHP are 820%, 1308%, and 360%, respectively, which are generally consistent with the measured values (810%, 1250%, and 335–340%, respectively).

The stretchability of the aerogel-derived porous elastomers significantly surpasses those of the previously reported stretchable aerogels without specially designed macroscopic structures (Fig. 3i and Supplementary Table 5). Moreover, the stretchability of our porous elastomers is among the highest in those of the reported stretchable foams and sponges based on inorganic building blocks[23,30,39] and various polymers including PEDOT:PSS[18], polyimide[20], polydimethylsiloxane (PDMS)[40–43], PU[44–49], poly(vinylidene fluoride) (PVDF)[50], and polyacrylate[51] (Fig. 3j and Supplementary Table 6).

Because of their compressed and folded porous structures, the uniaxially hot-pressed rGO/polymer elastomers show zero or low Poisson's ratios ($v$) over a wide range of tensile strains during stretching (Fig. 3a and Supplementary Fig. 30). The Poisson's ratios of rGO/PPy/PUF1-UHP are zero and 0–0.016 with tensile strains in the range of 0–100% and 100–800%, respectively, the values of which were much lower than those of rGO/PPy/PUF1 without hot pressing (0.305–0.316) (Supplementary Fig. 31). For rGO/PPy/PUF2-UHP, the Poisson's ratios are zero and 0–0.015 with tensile strains in the range of 0–200% and 200–1200%, respectively. For rGO/PPy/PU-UHP, the Poisson's ratios are zero and 0–0.014 with tensile strains in the range of 0–100% and 100–400%, respectively, the values of which were much lower than those of rGO/PPy/PU (0.268–0.312) (Supplementary Fig. 31). It should be noted that the biaxially and triaxially hot-pressed rGO/polymer elastomers exhibit negative Poisson's ratios during stretching because of their reentrant porous structures. The Poisson's ratios of rGO/PPy/PUF1-BHP in $x$ and $y$ directions are in the range of −(0.072–0.174) and −(0.032–0.120), respectively, while those of rGO/PPy/PU-BHP are in the range of −(0.091–0.110) and −(0.059–0.106), respectively (Fig. 3b, k and Supplementary Fig. 32). Besides, the Poisson's ratios of rGO/PPy/PUF1-THP in $x$, $y$, and $z$ directions are in the range of −(0.042–0.158), −(0.058–0.150), and −(0.039–0.118), respectively (Supplementary Fig. 33), while those of rGO/PPy/PU-THP are in the range of −(0.100–0.120), −(0.095–0.096), and −(0.067–0.094), respectively (Supplementary Fig. 34).

Negative Poisson's ratios of metamaterials can be obtained via many methods[8,29,32–36,52,53]. For example, shape-memory PU metaaerogels with negative Poisson's ratios can be obtained by using a suitable mold with a metastructure[52]. Metamaterials with negative Poisson's ratios can also be achieved by constructing reentrant 3D origami structures[53]. Compared with other methods, hot pressing is a relatively simple method to achieve negative Poisson's ratios. Metamaterials with negative Poisson's ratios show potential applications in soft robot, wave propagation manipulation, minimally invasive medical devices, etc.[52,53].

Moreover, the hot-pressed rGO/polymer elastomers can be reversibly compressed 80% in $z$ direction, showing high compressibility and elasticity (Supplementary Figs. 35 and 36). The compressive stresses of the elastomers have been significantly enhanced by hot pressing (Supplementary Figs. 35–37). The compressed skeletons with smaller pores, higher bulk densities, and lower porosities contribute to the higher compressive strength of the hot-pressed elastomers. Furthermore, the hot-pressed rGO/polymer elastomers exhibit high bendability (Supplementary Fig. 38). The high compressibility, bendability, and elasticity of the hot-pressed elastomers are probably attributed to the reversible deformability of PUF1, PUF2, PVA, and MF as well as the synergistic effect of rGO and the flexible polymers (Supplementary Figs. 26 and 39).

There is a recently published work on carbon tube aerogels with enhanced elasticity obtained by hot pressing[54]. In this published work, uniaxial hot pressing (1200 °C for 2 h) was applied to obtain SiC nanowire aerogels with a laminar structure, which also showed enhanced bulk densities and compressive strengths. The obtained carbon tube aerogels possessed near-zero Poisson's ratios upon compression, which are different from those of our porous elastomers that exhibit low or negative Poisson's ratios upon stretching.

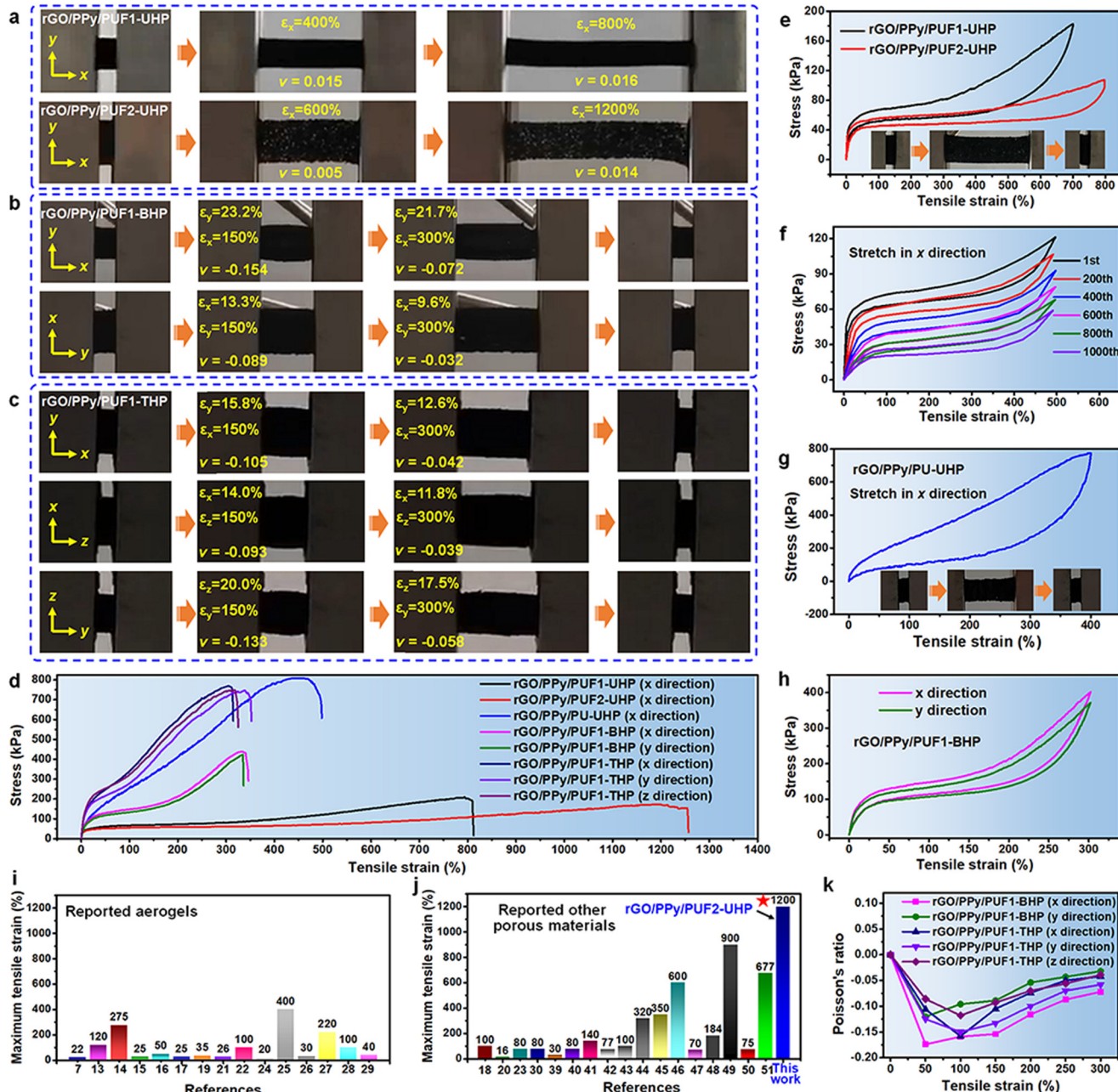

**Fig. 3 | Stretchability and Poisson's ratios of the hot-pressed rGO/polymer nanocomposite elastomers. a** Photographs of rGO/PPy/PUF1-UHP and rGO/PPy/PUF2-UHP upon stretching. **b** Photographs of rGO/PPy/PUF1-BHP upon stretching in different directions. **c** Photographs of rGO/PPy/PUF1-THP upon stretching in different directions. Poisson's ratios ($v$) of the elastomers were indicated in (**a**–**c**). **d** Stress-strain curves of the tensile tests on typical elastomers. **e** Stress-strain curves of the tensile tests in $x$ direction on uniaxially hot-pressed elastomers. Inset: photographs of rGO/PPy/PUF1-UHP during the tensile test. **f** Stress-strain curves of

rGO/PPy/PUF1-UHP for 1000 stretching-releasing cycles with 500% tensile strain in $x$ direction. Stress-strain curves of the tensile tests on (**g**) rGO/PPy/PU-UHP and (**h**) rGO/PPy/PUF1-BHP. The inset in (**g**) showed photographs of rGO/PPy/PU-UHP during the tensile test. Comparison of the stretchability of our porous elastomers and the reported stretchable (**i**) aerogels and (**j**) other porous materials. **k** Poisson's ratios of typical elastomers upon stretching with different tensile strains. Source data are provided as a Source Data file.

The in situ morphology observation of the hot-pressed rGO/polymer elastomers during stretching and compression was performed to investigate the deformation mechanism of the porous structures. The pore size became larger and the folded pore walls became unfolded along $x$ direction for the uniaxially hot-pressed elastomers (rGO/PPy/PUF1-UHP and rGO/PPy/PUF2-UHP) upon stretching in $x$ direction within 0–800% tensile strains (Fig. 4a and Supplementary Fig. 40). The morphologies of the hot-pressed elastomers after stretching are similar to those of the pristine aerogels without hot pressing. In the case of the biaxially hot-pressed elastomer

(rGO/PPy/PUF1-BHP), the reentrant pores became unfolded and the pore size became larger along both $x$ and $y$ directions upon stretching in $x$ or $y$ direction within 0–250% tensile strains (Fig. 4b). It is noteworthy that the pores moved away from the center along $y$ direction upon stretching in $x$ direction (Fig. 4b), confirming the negative Poisson's ratios of the biaxially hot-pressed rGO/polymer elastomers in terms of microstructure variations. Besides, it was observed that the pores of the hot-pressed elastomers were further compressed and the pore size became smaller upon compression in $z$ direction (Supplementary Fig. 41).

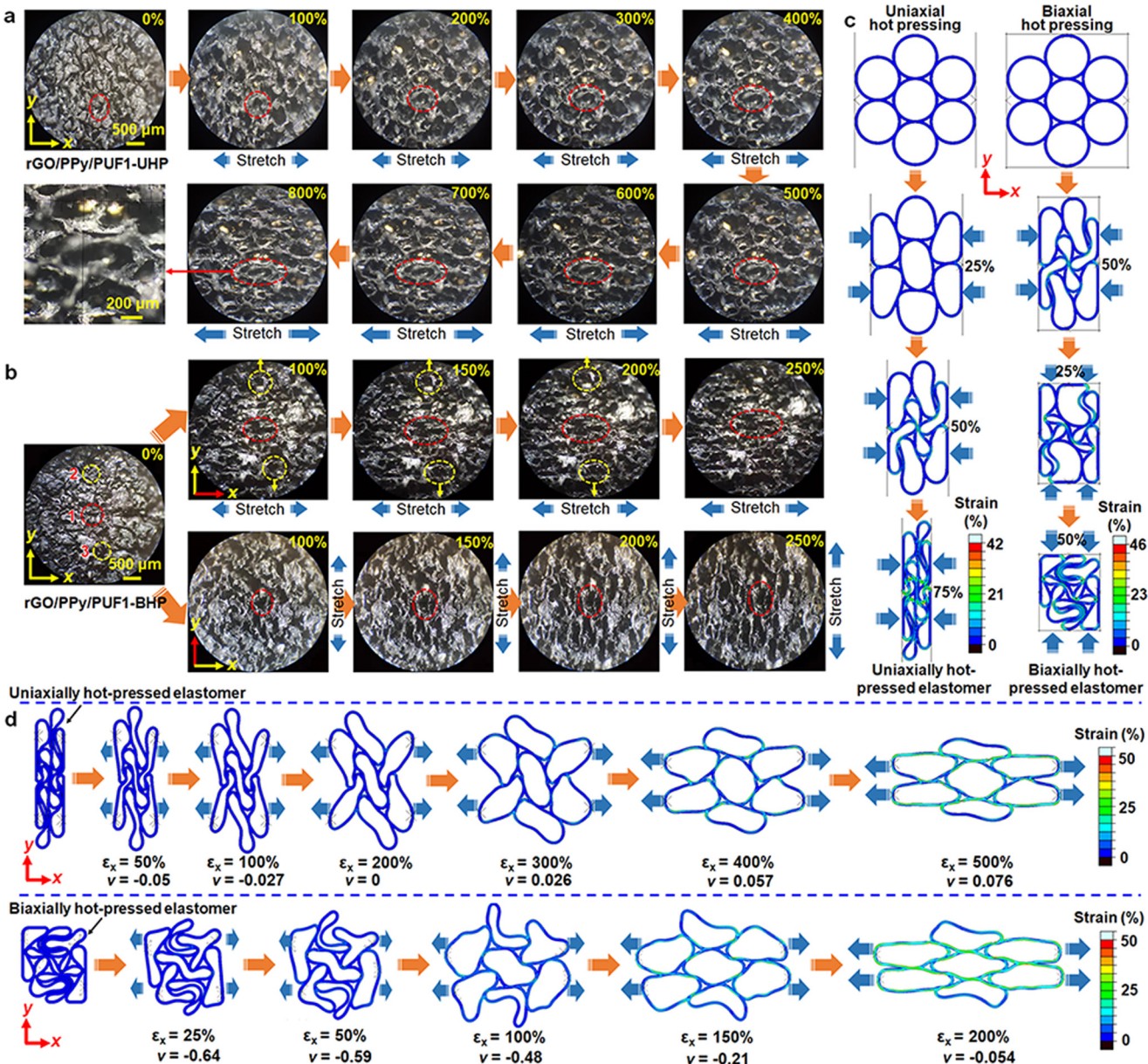

**Fig. 4 | In situ morphologies and FEA mechanical simulations of the porous rGO/polymer nanocomposite elastomers during hot pressing and stretching.** In situ optical microscope images of (**a**) rGO/PPy/PUF1-UHP and (**b**) rGO/PPy/PUF1-BHP upon stretching with tensile strains in the range of 0–800% and 0–250%, respectively. **c** FEA simulations of the structure variation and strain distribution of the rGO/polymer nanocomposite elastomers during uniaxial and biaxial hot pressing. **d** FEA simulations of the structure variation, strain distribution, and Poisson's ratios of the uniaxially and biaxially hot-pressed rGO/polymer nanocomposite elastomers during stretching.

As presented in the in situ optical microscope images of the hot-pressed rGO/polymer elastomers during stretching (Fig. 4a, b and Supplementary Fig. 40), the tensile deformation of the hot-pressed porous elastomers can be divided into three processes. The first deformation process is in the strain range of ~0–5%, during which many connected neighboring pore walls will be disconnected by overcoming the intramolecular or intermolecular interactions (such as hydrogen bonds). In this process, the stress increases significantly with the increase of tensile strain (Fig. 3d–h). The second deformation process is in the strain range of ~5% to $(1/(1-\varepsilon_0) - 1) \times 100\%$, where $\varepsilon_0$ is the compressive strain during hot pressing. For rGO/PPy/PUF1-UHP, $\varepsilon_0 = 75\%$ and $(1/(1-\varepsilon_0) - 1) \times 100\% = 300\%$. In this process, the folded and reentrant pores become unfolded and are gradually stretched to nearly their original shapes before hot pressing. This unfolding process can undergo large tensile strains. The strain of the third

deformation process is approximately larger than $(1/(1-\varepsilon_0) - 1) \times 100\%$. In this process, the pores are further stretched along the direction of tensile strain. The deformability in this process is determined by the stretchability of pristine rGO/polymer aerogels. The high deformability of the folded and reentrant microstructures during the second and third deformation processes mainly contribute to the high stretchability of the hot-pressed porous rGO/polymer elastomers. The high deformability of the polymers (such as PUF, PU, and MF) and the good combination of the polymers and rGO also contribute to their high stretchability.

In order to further understand the deformation mechanism of the porous microstructures of the hot-pressed rGO/polymer elastomers, the mechanical simulations during hot pressing and stretching were performed via finite element analysis (FEA) (Fig. 4c, d and Supplementary Figs. 42, 43). Since strain release during hot pressing is not

obvious (Supplementary Fig. 16), the FEA mechanical simulations are performed based on simplified hot-pressing and stretching processes without considering the strain release during the softening process. In the case of hot-pressing simulations, the pores of the elastomers were compressed and the pore walls were folded gradually upon uniaxial hot pressing (Fig. 4c and Supplementary Movie 8). Reentrant porous structures of the elastomers were formed after biaxial hot pressing (Fig. 4c and Supplementary Movie 9). The simulated folded and reentrant structures after uniaxial and biaxial hot pressing are generally consistent with the morphologies of the corresponding hot-pressed rGO/polymer elastomers as presented in Fig. 2 and Supplementary Figs. 9–11.

In the case of stretching simulations, the pore size became larger and the folded pore walls became unfolded along $x$ direction for the uniaxially hot-pressed elastomer upon stretching in $x$ direction (Fig. 4d). According to the simulation, the calculated Poisson's ratios of the uniaxially hot-pressed elastomer at tensile strains of 100%, 200%, 300%, and 400% are −0.027, 0, 0.026, and 0.057, respectively, indicating the zero or low Poisson's ratios of this kind of porous structures during stretching. For the biaxially hot-pressed elastomer, the reentrant porous structure became unfolded and the pore size became larger upon stretching in $x$ direction (Fig. 4d). The calculated Poisson's ratios of the simulated reentrant porous structure of the biaxially hot-pressed elastomer at tensile strains of 50%, 100%, 150%, and 200% are −0.59, −0.48, −0.21, and −0.054, respectively, confirming the negative Poisson's ratios via simulation. The simulated microstructure variations of the uniaxially and biaxially hot-pressed elastomers during stretching are generally consistent with the morphology variations obtained by in situ observation (Fig. 4a, b and Supplementary Fig. 40). More importantly, this simulation result verifies the versatility of the triaxial and biaxial hot-pressing strategies for constructing folded and reentrant porous structures with high stretchability and low or negative Poisson's ratios. Highly compressible aerogels with positive Poisson's ratios can be converted into highly stretchable porous elastomers with negative Poisson's ratios via these hot-pressing strategies.

## Sensing properties

Benefiting from their high stretchability, high compressibility, and high elasticity, the hot-pressed rGO/polymer elastomers can be used for ultrabroad-range-response strain and pressure sensors, which can't be achieved by traditional porous materials. The introduction of rGO and PPy makes the rGO/polymer elastomers electrically conductive, allowing the resultant elastomer-based strain/pressure sensors to be able to work in a resistive mode (Fig. 5a). The resistance of all the rGO/polymer elastomers increased with the increase of tensile strain and decreased with the increase of compressive strain (or pressure) (Fig. 5b, c). As shown in Fig. 4a, b, the pore size of the hot-pressed elastomers became larger and the amount of contact points of the neighboring pore walls decreased upon stretching, which resulted in the decreased conductive paths and increased resistance. On the contrary, the pores of the hot-pressed elastomers were compressed and more contact points were produced upon compression (Supplementary Fig. 41), which led to the increased conductive paths and decreased resistance.

The maximum detectable tensile strains of the strain sensors based on rGO/PPy/PUF1-UHP, rGO/PPy/PUF2-UHP, rGO/PPy/PU-UHP, rGO/PVA/MF-UHP, rGO/PPy/PUF1-BHP, and rGO/PPy/PUF1-THP reach 800%, 1200%, 450%, 100%, 300%, and 300%, respectively (Fig. 5b and Supplementary Fig. 44). The maximum detectable pressures of the pressure sensors based on rGO/PPy/PUF1-UHP, rGO/PPy/PUF2-UHP, rGO/PPy/PU-UHP, rGO/PPy/PUF1-BHP, and rGO/PPy/PUF1-THP reach 4.7, 4.4, 2.8, 8.2, and 9.5 MPa, respectively (Fig. 5c). The response of the rGO/PPy/PUF1-UHP-based strain sensor remained nearly unchanged during stretching-releasing with 400% strain for 1000 cycles, indicating its high durability and fatigue resistance (Fig. 5d). This makes them

suitable for broad-range-response strain sensors for detecting tensile strains. The maximum detectable tensile strain (1200%) of the strain sensor based on rGO/PPy/PUF2-UHP is among the largest in those of the previously reported strain sensors based on stretchable conductive aerogels, foams, and sponges (Fig. 5e and Supplementary Tables 5 and 6)[6,18–25,41–44,46,49]. Besides, the rGO/PPy/PUF1-UHP-based pressure sensor can withstand compression-decompression with 80% strain for 1000 cycles (Supplementary Fig. 45). However, the compressive stress still decreased after repeated compression-decompression (Supplementary Fig. 35). The stability of the pressure sensors needs to be further enhanced. The measured lower detection limit of the rGO/PPy/PUF1-UHP-based pressure sensor is 1.5 kPa. Therefore, the rGO/PPy/PUF1-UHP-based pressure sensor shows a broad detection range of 1.5 kPa–4.7 MPa.

Because of their ultrabroad detection ranges, the strain/pressure sensors based on the hot-pressed rGO/polymer elastomers can be used as strain- or pressure-sensitive wearable electronics. Finger and wrist bending and muscular movement could be monitored in real time by attaching the rGO/PPy/PUF1-UHP-based sensor on the surfaces of a finger, wrist, and muscle, respectively, demonstrating its application potentials in monitoring human body motions (Fig. 5f and Supplementary Fig. 46). More importantly, the hot-pressed elastomer-based strain sensors can be used for monitoring large tensile strains of elastic rope during bungee jumping, chest developer during exercise, coil spring in some mechanical equipment during stretching, soft robot, etc. For example, the rGO/PPy/PUF1-UHP-based sensor could monitor the large tensile strain (0–600%) of a balloon that was blown up by attaching it on the surface of the balloon (Fig. 5g). Besides, it could monitor the large tensile strain (0–300%) of a chest developer during exercise (Fig. 5h and Supplementary Movie 10). Furthermore, we demonstrated that the hot-pressed elastomer-based strain sensors showed potential applications in robots and prostheses. Five rGO/PPy/PUF1-UHP-based strain sensors were fixed on five fingers of a bionic hand. The strain sensor would be stretched when the finger was bent, resulting in the increased resistance of the sensor. Different gestures of the bionic hand could be monitored in real time by recording the response of each strain sensor on the fingers (Fig. 5i).

## Smart thermal management

Because of their porous structures, the hot-pressed elastomers can be used as thermal insulators for thermal management. The thermal conductivities of rGO/PPy/PUF1-UHP and rGO/PPy/PU-UHP at room temperature and ambient pressure are 0.034 and 0.029 W m$^{-1}$ K$^{-1}$, respectively, the values of which are lower than those of some commercial thermal insulation materials such as mineral wool and comparable to those of extruded polystyrene and polyurethane foam, indicating that they are thermal insulators[5]. Traditional thermal insulation materials usually show fixed thermal insulation performance. The pore sizes of traditional thermal insulation materials usually can't be reversibly tuned by stretching. However, the pore sizes of the hot-pressed rGO/polymer elastomers are highly tunable over a wide range simply via stretching with different strains (Fig. 4a, b and Supplementary Fig. 40). Since the hot-pressed elastomers show high stretchability, high elasticity, and reversibly tunable pore sizes, they are expected to be able to achieve reversibly tunable thermal insulation for smart thermal management via stretching.

The thermal insulation performances of the hot-pressed rGO/polymer elastomers under different tensile strains were investigated by infrared thermography. The infrared thermal images of rGO/PPy/PUF1-UHP (2 mm thick) with different tensile strains on a hot plate (51 °C) were observed in real time (Fig. 6a). The temperature of the top surface of the elastomer increased with the increase of tensile strains in spite of the larger size of the stretched elastomer. The top surface temperature of pristine rGO/PPy/PUF1-UHP stabilized at 36.1 °C, the value of which is lower than those of the stretched elastomer with

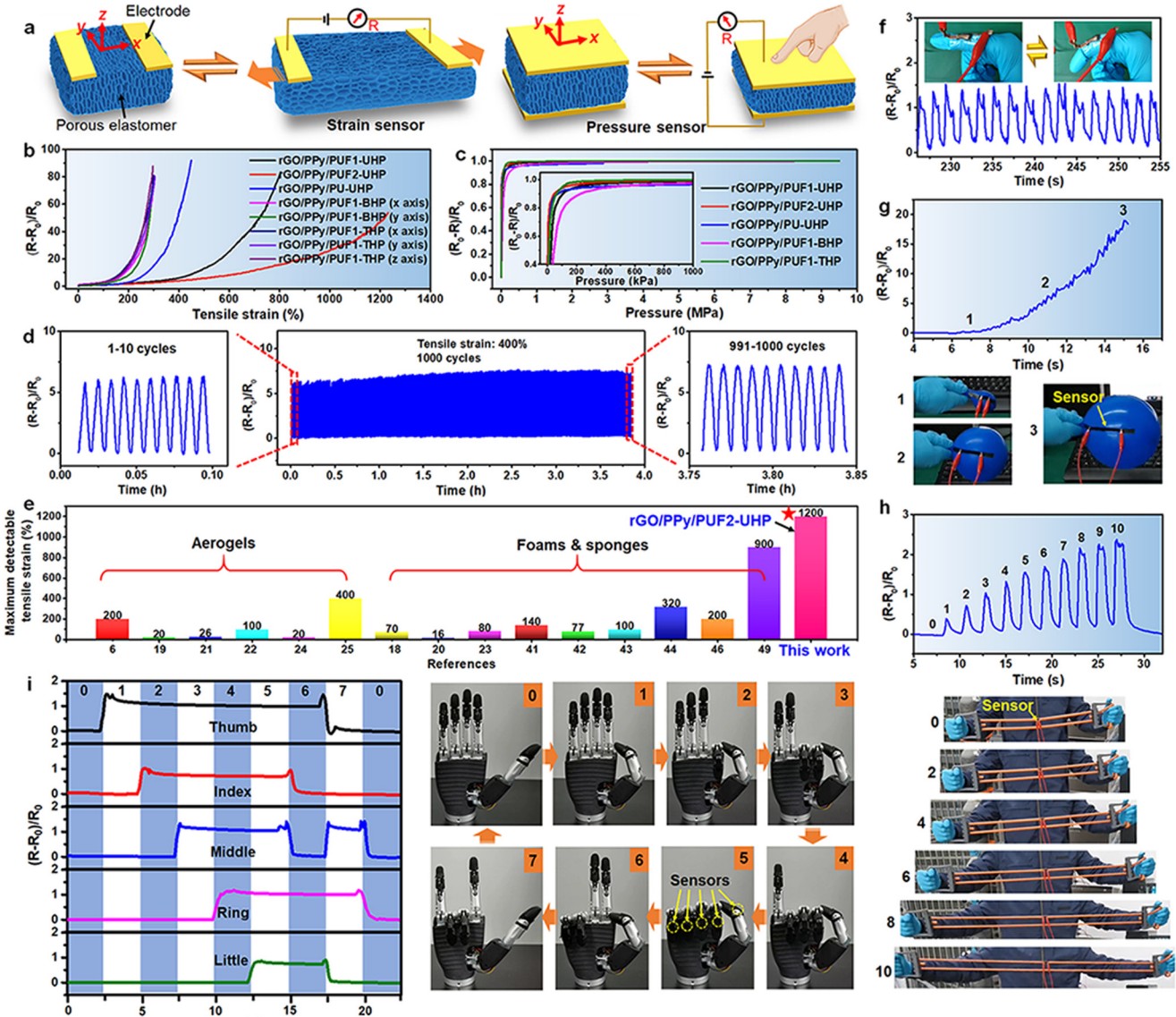

**Fig. 5 | Sensing performances of the strain and pressure sensors based on the hot-pressed rGO/polymer nanocomposite elastomers. a** Schematic of the strain/pressure sensors based on the stretchable elastomers. **b** Relative resistance changes versus tensile strain for the strain sensors based on different elastomers. **c** Relative resistance changes versus pressure (in $z$ direction) for the pressure sensors based on different elastomers. **d** Stretching durability test of the rGO/PPy/PUF1-UHP-based strain sensor with 400% tensile strain in $x$ direction for 1000 cycles. **e** Comparison of the maximum detectable tensile strains of our stretchable porous elastomer-based strain sensors and the previously reported strain sensors based on aerogels and other porous materials. **f** Monitoring finger bending by the rGO/PPy/PUF1-UHP-based sensor. Monitoring tensile strains of (**g**) a balloon that was blown up and (**h**) a chest developer during exercise by the rGO/PPy/PUF1-UHP-based sensor. **i** Real-time responses of the strain sensors fixed on five fingers of a bionic hand with different gestures. Source data are provided as a Source Data file.

80–250% tensile strain (Fig. 6b). The larger pore size and lower apparent density under a larger tensile strain may result in the higher thermal conductivities of gas and radiation, leading to the lower thermal insulation of the hot-pressed elastomer with larger tensile strains (Fig. 6c and Supplementary Note 5)[5,55,56]. As expected, the hot-pressed rGO/polymer elastomers exhibit tunable thermal insulation and can be applied for smart thermal management, which is achieved by regulating the porous microstructures simply via stretching.

In addition, the thermal stability of the hot-pressed elastomers was studied (Supplementary Figs. 47–49 and Supplementary Note 6). It is found that the mechanical properties of typical hot-pressed porous elastomers remain nearly unchanged after heat treatment at temperatures below 100 °C. Since the thermal insulation and management demonstrations were carried out at the temperature of 51 °C, the materials are thermally stable at this applied condition.

## Smart EMI shielding

Smart EMI shielding materials with reversibly tunable EMI shielding performances are attractive for next-generation EMI shielding devices. However, traditional EMI shielding materials usually exhibit fixed EMI shielding performance[9,57,58]. Here, reversibly tunable EMI shielding has been achieved by the highly stretchable hot-pressed rGO/polymer elastomers simply via stretching with different strains.

The total shielding effectiveness (SE$_T$) and absorption effectiveness (SE$_A$) of the uniaxially hot-pressed elastomer rGO/PPy/PUF1-UHP in $z$ direction obviously decreased with the increase of tensile strain (Fig. 6d, e). The SE$_T$ of pristine rGO/PPy/PUF1-UHP is -40 dB, the value of which is higher than the requirement (>20 dB) of the standard EMI shielding. Therefore, pristine rGO/PPy/PUF1-UHP can be regarded as a kind of EMI shielding materials. By contrast, the SE$_T$ of rGO/PPy/PUF1-UHP at 100%, 200%, and 300% tensile strains are only -17.3, 9.8, and

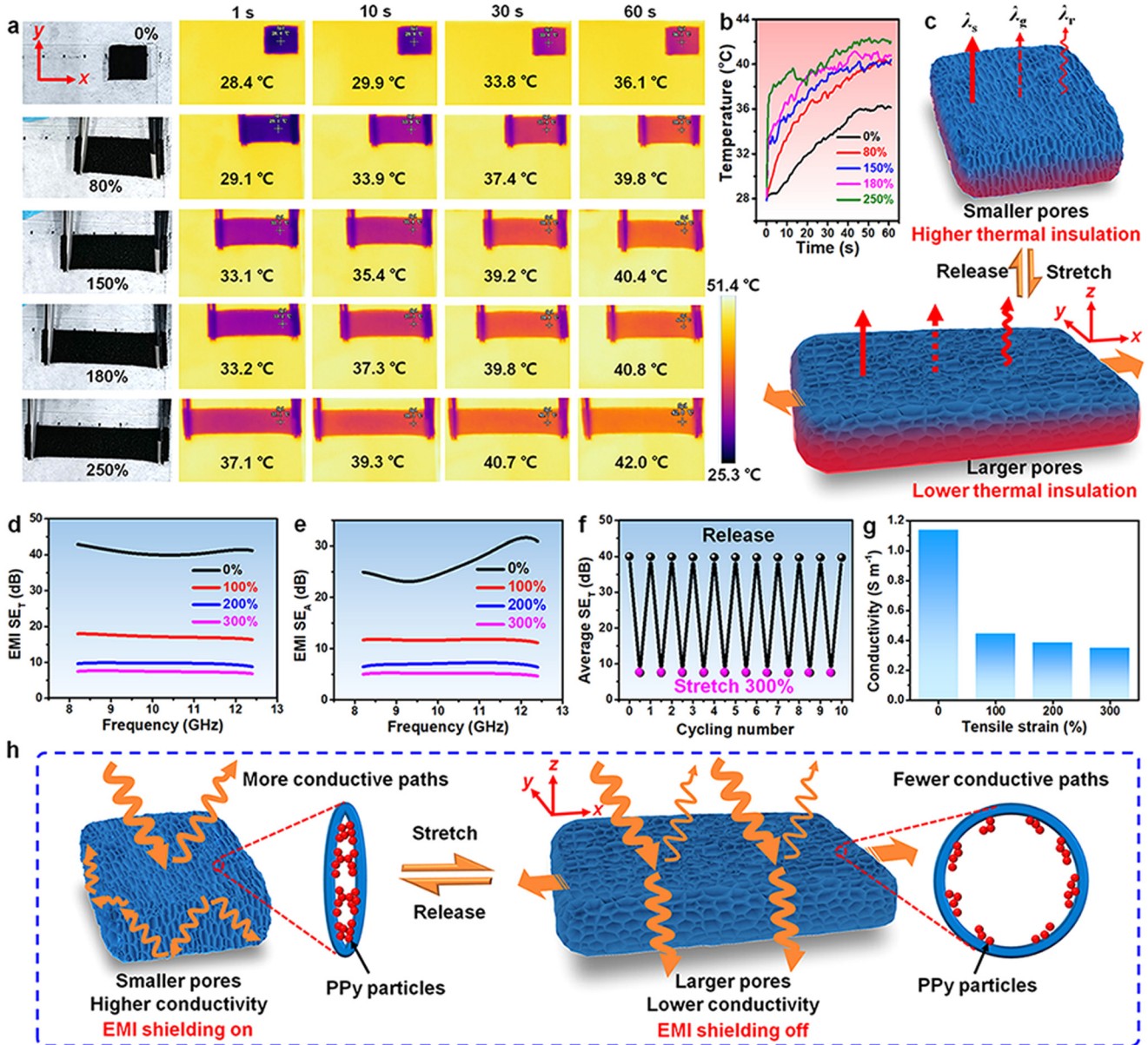

**Fig. 6 | Smart thermal management and EMI shielding performances of the hot-pressed stretchable rGO/polymer nanocomposite elastomers. a** Real-time infrared thermal images and (**b**) temperatures of pristine and stretched rGO/PPy/PUF1-UHP (2 mm thick) with different tensile strains on a hot plate. **c** Mechanism of smart thermal management of the stretchable elastomers achieved by stretching. **d** SE$_T$ of rGO/PPy/PUF1-UHP in $z$ direction with different tensile strains. **e** SE$_A$ of rGO/PPy/PUF1-UHP in $z$ direction with different tensile strains. **f** EMI shielding cycling performance of rGO/PPy/PUF1-UHP for 10 stretching-releasing cycles with 300% tensile strain. **g** Electrical conductivities of rGO/PPy/PUF1-UHP before and after stretching with different tensile strains. **h** Mechanism of smart EMI shielding of the stretchable elastomers achieved by stretching. Source data are provided as a Source Data file.

7.5 dB, respectively. As we can see, EM wave shielding for the hot-pressed rGO/polymer elastomer can be converted to EM wave transmission simply via stretching. The reversible switch between EM wave shielding and transmission of the hot-pressed elastomer can be achieved by repeatedly stretching and releasing the elastomer (Fig. 6f).

EMI shielding is influenced by the reflection, multiple reflections, and absorption of the EM wave[58,59]. Since the pore size of the stretched rGO/polymer elastomers becomes larger, there are fewer contact points between neighboring conductive rGO nanosheets and PPy particles, resulting in fewer electron transport paths. As a result, the electrical conductivities of the hot-pressed rGO/polymer elastomers became lower upon stretching (Fig. 6g). In addition, the larger pores of the elastomers may result in lower permittivity[59]. Therefore, the impedance matching became better and the dielectric loss and attenuation of EM waves inside the elastomer were decreased for the stretched rGO/

polymer elastomer, leading to the reduced reflection and increased transmission of EM waves (Fig. 6h). This kind of highly stretchable porous elastomers with reversibly tunable EMI shielding performances show potential applications in smart EMI shielding devices.

The major novelty and significance of this work are summarized as follows. (1) Various highly compressible aerogels with positive Poisson's ratios can be converted into highly stretchable meta-elastomers with near-zero or negative Poisson's ratios via the versatile uniaxial/biaxial/triaxial hot-pressing strategies. (2) The stretchability of the meta-elastomers significantly surpasses those of the reported stretchable aerogels without specially designed macroscopic structures and is among the highest in those of the reported stretchable porous materials. (3) The mechanism of shape fixing during hot pressing and high stretchability of this kind of meta-elastomers are revealed, which will provide important theoretical support for

designing highly stretchable metamaterials. (4) The meta-elastomers can achieve high biaxial (or triaxial) stretchability and negative Poisson's ratios during stretching in different directions, which is difficult to be achieved by traditional negative-Poisson-ratio porous metamaterials. (5) The maximum detectable tensile strain (1200%) of the meta-elastomer-based strain sensor is among the largest in those of the reported strain sensors based on aerogels, foams, and sponges. (6) The reversibly tunable thermal insulation and EMI shielding performances of the meta-elastomers overcome the limitations of the fixed thermal insulation and EMI shielding of traditional porous materials.

The limitation of these hot-pressing strategies in this work is that highly stretchable elastomers are difficult to be obtained by hot pressing of the porous materials without high compressibility and elasticity. Besides, the electrical conductivities of these hot-pressed stretchable elastomers need to be further improved to broaden their potential applications in flexible electronics.

## Discussion

Several types of intrinsically highly stretchable conductive rGO/polymer nanocomposite elastomers with low or negative Poisson's ratios have been prepared via uniaxial, biaxial, and triaxial hot-pressing strategies. Highly compressible aerogels with positive Poisson's ratios can be converted into highly stretchable meta-elastomers with zero or negative Poisson's ratios via these hot-pressing strategies. The uniaxially hot-pressed elastomers possessed compressed and folded porous structures and exhibited zero or low Poisson's ratios, high stretchability up to 1200% strain, high compressibility, and high elasticity. The biaxially hot-pressed meta-elastomers possessed reentrant porous structures and showed high biaxial stretchability and negative Poisson's ratios. Furthermore, the meta-elastomers combining reentrant porous structures, high triaxial stretchability, and negative Poisson's ratios in different directions were achieved by triaxial hot pressing. The resulting elastomers can be applied for ultrabroad-range-response wearable strain (0–1200%) and pressure (0–9.5 MPa) sensors. In addition, they can be used for reversibly tunable thermal management and EMI shielding, which are achieved by regulating the porous microstructures simply via stretching. We anticipate that these versatile strategies can be utilized to develop various highly stretchable porous materials with low or negative Poisson's ratios, and may endow them with new properties and application possibilities in flexible electronics, thermal management, EMI shielding, energy storage, etc.

## Methods
### Materials
Graphene oxide (GO, 4 mg/ml) was purchased from Institute of Coal Chemistry, Chinese Academy of Science. Ethylenediamine (EDA, AR) was purchased from Sinopharm Chemical Reagent Co., Ltd. Pyrrole (99%) and Iron chloride (FeCl$_3$, 98%) was purchased from Adamas. Polyvinyl alcohol (PVA, $n = 1750 \pm 50$) was purchased from Tokyo Chemical Industry Co., Ltd. Polyurethane (PU) dispersion (Dispercoll U42, 50 wt% in water) was purchased from Covestro. 3-Aminopropyltriethoxysilane (APTES, 98%) was purchased from Energy Chemical. The PU foams (PUF1 and PUF2) were purchased from Jiangsu Huayi Foam Technology Co., Ltd and Yuechang Foam. The melamine foam (MF) was provided by Zhejiang Yadina New Material Technology Co., Ltd.

### Preparation of rGO/PUF-based aerogel
The rGO/PPy/PUF composite aerogels were prepared by in situ formation of rGO aerogel in PUF followed by the deposition of PPy. In total, 0.032 ml EDA (AR) was mixed with 4 ml GO (4 mg ml$^{-1}$) under continuous stirring for 5 min. PUF1 or PUF2 strips were completely immersed into the above mixture followed by repeated squeezing. The PUF strips impregnated with the GO sol were then sealed and heated at

90 °C for 6 h to afford rGO/PUF composite hydrogels. After ambient pressure drying (APD) at 80 °C for at least 2 h, the rGO/PUF composite aerogels were obtained.

PPy was then deposited in the aerogels via the following method. In total, 0.2268 g of FeCl$_3$ (98%) was dissolved in 8 ml of H$_2$O (DIW). In total, 0.066 ml of Pyrrole (99%) was mixed with the above FeCl$_3$ solution in an ice bath under stirring for 20–30 s. Then, the rGO/PUF composite aerogels were immersed into the above mixture and squeezed repeatedly. After polymerization at room temperature for 6 h, the rGO/PPy/PUF composite hydrogels were formed. The rGO/PPy/PUF composite aerogels (rGO/PPy/PUF1 and rGO/PPy/PUF2) were obtained after APD at 80 °C for at least 2 h. The preparation parameters of the rGO/PPy/PUF composite aerogels were presented in Supplementary Table 1.

### Preparation of rGO/PU-based aerogel
The rGO/PU-based aerogel rGO/PPy/PU was prepared via the following steps. As shown in Supplementary Table 2, 0.096 ml APTES was added into the mixture of 8 ml H$_2$O (DIW), 8 ml GO (4 mg ml$^{-1}$), and 1.28 ml PU dispersion (50 wt% in water) under continuous stirring for 1 min. After gelation at room temperature, the obtained hydrogel was sealed and aged at 80 °C for 6 h. The hydrogel was freeze-casted at −17 °C for at least 6 h followed by freeze drying for at least 12 h to afford the rGO/PU composite aerogel. PPy was then deposited in the rGO/PU aerogel to form an rGO/PPy/PU composite hydrogel via the same method as that for the deposition of PPy in the rGO/PUF composite aerogels except for the drying process. The rGO/PPy/PU composite aerogel was obtained by freeze drying, followed by heat treatment in vacuum at 120 °C for 1 h.

### Preparation of rGO/MF-based aerogel
The rGO/PVA/MF composite aerogel was prepared by in situ formation of rGO/PVA aerogel in MF. As shown in Supplementary Table 3, 4 ml GO (4 mg ml$^{-1}$), 0.16 ml PVA (0.05 g ml$^{-1}$), and 0.032 ml EDA (AR) were mixed and continuously stirred for 10 min. MF strips were completely immersed into the above mixture followed by repeated squeezing. The MF strips impregnated with the GO sol were then sealed and heated at 90 °C for 24 h to form rGO/PVA/MF composite hydrogels. The rGO/PVA/MF composite aerogels were obtained by freeze-casting at −17 °C for at least 6 h followed by freeze drying for at least 12 h. To further reduce the rGO, the rGO/PVA/MF composite aerogels were then thermally treated at 150 °C for 5 h in a tube furnace in N$_2$ atmosphere.

### Preparation of hot-pressed porous rGO/polymer composite elastomers
Hot pressing of the rGO/polymer nanocomposite aerogels was readily performed using a home-made apparatus (Fig. 1c and Supplementary Figs. 3–8). In the case of uniaxial hot pressing (Supplementary Figs. 3 and 6), a monolithic rGO/polymer aerogel was first sandwiched with two pieces of glass and then compressed with 66.7–87.5% strain (75% for rGO/PPy/PUF1 and rGO/PPy/PU, 87.5% for rGO/PPy/PUF2, and 66.7% for rGO/PVA/MF) in $x$ direction using other two pieces of glass. After that, they were fixed and heated at 120 °C or 140 °C. rGO/PVA/MF and rGO/PPy/PU were heated at 120 °C (heated for 5 h for rGO/PVA/MF and 0.5 h for rGO/PPy/PU) to afford rGO/PVA/MF-UHP and rGO/PPy/PU-UHP, respectively, while rGO/PPy/PUF1 and rGO/PPy/PUF2 were heated at 140 °C for 5 h to afford rGO/PPy/PUF1-UHP and rGO/PPy/PUF2-UHP, respectively.

In the case of biaxial hot pressing (Supplementary Figs. 4 and 7), a monolithic rGO/polymer aerogel was first compressed with 50% strain in $x$ direction using two pieces of glass and then compressed with 50% strain in $y$ direction using other two pieces of glass, followed by fixing and heat treatment. rGO/PPy/PUF1 was biaxially compressed and heated at 140 °C for 5 h to afford rGO/PPy/PUF1-BHP, while rGO/PPy/PU was biaxially compressed and heated at 120 °C for 0.5 h to afford

rGO/PPy/PU-BHP. In the case of triaxial hot pressing (Supplementary Figs. 5 and 8), a monolithic rGO/polymer aerogel was first compressed with 50% strain in $z$ direction and then compressed with 50% strain along both $x$ and $y$ directions, followed by fixing and heat treatment. rGO/PPy/PUF1 was triaxially compressed and heated at 140 °C for 5 h to afford rGO/PPy/PUF1-THP, while rGO/PPy/PU was triaxially compressed and heated at 120 °C for 0.5 h to afford rGO/PPy/PU-THP.

## Preparation of rGO/polymer composite elastomer-based sensors

As shown in Fig. 5a, the flexible strain sensors were constructed by attaching two copper electrodes on both ends of the hot-pressed rGO/polymer composite elastomers (coplanar electrode structure). For the flexible pressure sensors, the hot-pressed rGO/polymer composite elastomers were sandwiched by two copper electrodes (vertical electrode structure).

## Structural characterizations

The porosities of different materials were calculated by:

$$P = \left(1 - \frac{\rho_0}{\rho}\right) \times 100\% \tag{1}$$

where $P$ was porosity, $\rho_0$ was bulk density of the porous materials, and $\rho$ was skeletal density. The morphologies of different samples were observed by a scanning electron microscope (SEM, NOVA NANOSEM 450, FEI, USA), a transmission electron microscope (TEM, Tecnai F20, FEI, USA), and an optical microscope (6XB-PC, Shang Guang, PR China). The in situ morphologies of the hot-pressed elastomers during stretching and compression were observed by the same optical microscope. X-ray diffraction (XRD) patterns were recorded on an X-ray diffractometer (5–80°, 2° min$^{-1}$) (DX-2701BH, Haoyuan, PR China). Raman spectra were studied with a Raman spectrometer with an argon ion laser excitation wavelength of 532 nm at room temperature (LabRAM HR Evolution, HORIBA Scientific, Japan). Fourier transform infrared (FTIR) spectra in the range of 4000–400 cm$^{-1}$ were investigated using an FTIR spectroscope (FTIR, Nicolet iS20, Thermo Fisher, USA). X-ray photoelectron spectra (XPS) of different aerogels were measured via an X-ray photoelectron spectrometer (ESCALAB Xi +, Thermo Scientific, USA). The thermogravimetric (TG) and derivative thermogravimetric (DTG) curves were measured in air by a TG analyzer (TG 209F3, NETZSCH, Germany) from 30 to 800 °C with a heating rate of 10 K min$^{-1}$.

## Mechanical tests

Storage modulus, loss modulus, and tan delta of samples were measured by a thermomechanical analyzer (DMA Q800, TA Instruments, USA). Tensile and compressive stress-strain curves of different samples were measured by a micro-computer controlled electronic universal testing machine (LD23.104, Lishi (Shanghai) Instruments Co., Ltd., PR China). The stretching speed was fixed at 120 mm min$^{-1}$ for the cyclic stretching-releasing tests of the rGO/polymer elastomers and the strain sensors based on the elastomers. For the cyclic compression-decompression tests of the rGO/polymer elastomers and the pressure sensors based on the elastomers, the compression speed was fixed at 80 mm min$^{-1}$. For other stretching and compression tests, the speed was fixed at 40 mm min$^{-1}$. In the case of three-point bending tests, the speed was fixed at 10 mm min$^{-1}$ and the span of the fixture was kept at 13 mm.

## Mechanical simulations via FEA

FEA software ABAQUS was used to simulate the mechanical performances of the rGO/polymer composite elastomers. The simulations of the structure variation and stress/strain distributions of the elastomers during uniaxial (or biaxial) hot pressing and stretching were carried out via the explicit dynamic analysis method. A model containing

seven interconnected annuluses with the same size was used for the simulation. The annuluses were connected with each other at each contact point. Each annulus was deformable and elastic and possessed the same modulus. The elastomer was modeled by plane strain elements (CPE4R), while the compression plate was modeled by rigid bodies (R2D2). The density, elastic modulus, and Poisson's ratio of the annuluses used in simulation were 1 g cm$^{-3}$, 6 MPa, and 0.47, respectively.

For the simulation of uniaxial hot pressing of the elastomer, the annulus model was compressed 75% in $x$ direction, followed by the elimination of the internal stress to afford a fixed compressed structure. For the simulation of biaxial hot pressing of the elastomer, the annulus model was first compressed 50% in $x$ direction and then compressed 50% in $y$ direction, followed by the elimination of internal stress. In the case of stretching simulation, the compressed annulus model was stretched in $x$ direction with different tensile strains (0–500% for the uniaxially hot-pressed elastomer and 0–200% for the biaxially hot-pressed elastomer). The internal stress of the compressed annulus model was eliminated before the stretching simulation.

## Sensing tests

The tensile strain and pressure sensing properties of different samples were investigated by a micro-computer controlled electronic universal testing machine (LD23.104, Lishi (Shanghai) Instruments Co., Ltd., PR China) combined with a LCR meter (TH2840A, Tonghui Instrument Co., Ltd., PR China). The strain and pressure sensors were fixed between the top and bottom clamps (or plates) of the micro-computer controlled electronic universal testing machine and their electrodes were connected to a LCR meter. In addition, the rGO/polymer elastomer-based strain/pressure sensors were attached to a volunteer's body, a balloon, a chest developer, and fingers of a bionic hand to monitor different signals. The experiments were approved by the ethical committee of Tongji University. Xiaoyu Zhang provided written informed consent to participate in the body sensing tests.

## Thermal management tests

The hot-pressed elastomer rGO/PPy/PUF1-UHP (2 mm thick) with different tensile strains (in the range of 0–250%) was placed on the surface of a hot plate (JF-956, 51 °C, Dongguan Changan Jinfeng Electronic Tools Factory, PR China). The real-time infrared thermal images of the elastomer at different tensile strains were recorded using a thermal imager (626-L28, Fotric, PR China). Temperatures of the top surface of the elastomer under different tensile strains during 0–60 s were obtained from the video recorded by the thermal imager. The thermal conductivities were measured by the plate thermal flow method using a thermal conductivity tester (DRPL-III, Xiangtan Xiangyi instrument Co., Ltd, PR China).

## EMI shielding tests

The EMI SE (in $z$ direction) of rGO/PPy/PUF1-UHP with different tensile strains was measured by a rectangular waveguide (32117) using a 2-port network analyzer (3672B-S, Ceyear, PR China) in the frequency range of 8.2–12.4 GHz. The power coefficients of reflection ($R$), transmission ($T$), and absorption ($A$) were calculated from the scattering parameters of the reflection coefficient $S_{11}$ and the transmission coefficient $S_{21}$ based on the following equations:

$$R = \left|S_{11}\right|^2 \tag{2}$$

$$T = \left|S_{21}\right|^2 \tag{3}$$

$$A = 1 - (T + R) \tag{4}$$

The total EMI SE ($SE_T$), reflection effectiveness ($SE_R$), and absorption effectiveness ($SE_A$) were calculated by the following equations:

$$SE_T = 10 \log \frac{1}{T} \tag{5}$$

$$SE_R = 10 \log \left( \frac{1}{1-R} \right) \tag{6}$$

$$SE_A = 10 \log \left( \frac{1-R}{T} \right) \tag{7}$$

$$SE_T = SE_R + SE_A \tag{8}$$

The conductivities ($\sigma$) were calculated via the following equation:

$$\sigma = \frac{L}{R \times S} \tag{9}$$

where $R$ is the resistance, $L$ is the length, $S$ is the cross-sectional area of the sample.

## Data availability

The data generated in this study are provided in the Manuscript, Supplementary Information, and Source Data file. All other data that support the findings of this study are available from the corresponding author upon a request. Source data are provided with this paper.

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

## Acknowledgements

This work was supported by the National Key Research and Development Program of China (2022YFB3203500 to J.H.), the Shanghai Social Development Science and Technology Project (20dz1201800 to G.W.), the National Natural Science Foundation of China (62074111 to J.H., 62374113 to G.Z.), the Science & Technology Foundation of Shanghai (19JC1412402 and 20JC1415600 to J.H.), the Innovation Program of Shanghai Municipal Education Commission (2021-01-07-00-07-E00096 to J.H.), and the Fundamental Research Funds for the Central Universities (22120230224 to G.Z.).

## Author contributions

X.Z. prepared and synthesized the samples, characterized the structures, tested the mechanical, sensing, thermal management, and EMI shielding performances, processed data and wrote the experimental section and the structure analysis of the manuscript. Q.S. collected relevant references. Q.S. and X.L. prepared the schematic diagrams. P.G. and Z.H. checked and confirmed the rationality of the research. X.Y., M.L., and Z.S. checked the language and format of the manuscript. J.H. was responsible for the project administration and funding. G.W. was responsible for the funding. G.Z. conceived and designed the research, processed data, wrote and revised the manuscript, supervised the research, and was responsible for the project administration and funding.

## Competing interests

The authors declare no competing interests.
