## [Peer Review File · Nature Communications]

Stretchable and negative-Poisson-ratio porous metamaterialsREVIEWER COMMENTS

Reviewer #1 (Remarks to the Author):

This article proposes a stretchable auxetic material which is presented by some interesting applications. I found the article interesting however our fundamental understanding on the material based on the provided chemical/physical characterizations is limited. Below I summarize my concerns/comments.

1 - This material is not an "aerogel". Although you started with a freeze-dried material but at the end you work with a compressed material which is no longer has any aerogel characteristics. Therefore, comparing this material with aerogel materials is not apple-to-apple comparison. I think the material should be classified as an elastomer, not an aerogel.

2 - The material was not characterized in terms of density, porosity, pore size and surface area, before and after hot pressing process. Just because you freeze-dry your material, it is not guaranteed you end up to a high surface area material. So, these characterizations should be conducted.

3 - What is the mechanism for shape fixing at high temperature? Is it a reversible shape fixing process if the material is heated back to high temperature? Is it dominated by the PU foam matrix? What the glass transition looks like for the material? It seems there might be multiple transitions.

4 - In figure 1, a chemical structure was shown. However, no data is provided to support the structure. How do you know if this is what you have instead of just an intertwined multi-strand network?

In my opinion, the article is dealing with an interesting material but is not ready for publication at this condition. I recommend a major revision.

Reviewer #2 (Remarks to the Author):

This manuscript reported the preparation and properties of a class of highly stretchable rGO/PPy/PU composite aerogels with different Poisson ratio. The aerogels showed large tensile deformations, which is an advance when comparing with other aerogels. However, to improve the manuscript, there are more works should be down in the characterization of the aerogels, the explanation and discussion of the mechanical properties, and also the functional properties. The followings are my comments and suggestions.

1.As a composite aerogel, all the compositions and the interfaces between them in the aerogel are responsible for the mechanical property and functions. The current results in the characterizations of the structures of the aerogels are not sufficient. For examples, the interface between the polymer and rGO is not clearly, the morphology and crystal size of the rGO are not mentioned. These structure characters are very important for the mechanical and functional properties.

2.Densities and porosities of the composite aerogels are missed. These parameters are very important for the determinations and evaluation of their mechanical properties and thermal insulation performance.

3.As the authors claim that "Softening and plastic deformations of the polymers (PUF, PU, PVA, or MF) in the skeletons of the rGO/polymer aerogels would occur when the aerogels were compressed and heated at 120-140 °C", strains in the structure might be released during the softening process. Therefore, simply using FEA results to reflect the strain distribution (Figure 4c) in the microstructure of the aerogel is not accurate.

4.To explain the reasons for the high stretchability, the authors showed three main reasons. "First, the flexibility of the polymers (PUF, PU, PVA, or MF) endows the rGO/polymer networks with excellent reversible deformability, which can allow the aerogels to be compressed with a large

strain without fracture during hot pressing. Second, the compressed rGO/polymer networks can be fixed without springing back after being cooled to room temperature. Third, the obtained folded or reentrant porous rGO/polymer structures after hot pressing are highly stretchable." One can find that this kind of explanation is used phenomena to explain phenomena. The stretchability is the mechanical property of the composite aerogel. The author should analyze the deformation process from observe the microstructure characters of the aerogels themselves and microstructure changes during tensile deformation.

5. What does the "intrinsically stretchability" mean? For most inorganic materials, large strain deformation may not their intrinsic properties. The stretchability of most graphene based aerogels, and nanofibers based aerogels are mostly from their nanometer scale building units. Moreover, the deformation ability of polymeric and inorganic materials is different, their mechanical properties can not be simply comparee.

6. A high accuracy is very important for a strain or pressure sensor. The demands for strain sensor and stress sensor might be different. A material that is suitable for strain sensor may not be suitable for stress sensor. During the 1000 stretching-releasing cycles, it could be seen that the stress decrease significantly. One could question that would the present aerogel be suitable for sensors. Moreover, for a very large strain, like 500% or 1000% tensile strain, would it need a sensor to detect? Is there any real application situations?

For the thermal insulation performance of the aerogel, it could be seen that the surface of the aerogel reached 42.4 °C in 30s when the substrate is only 50 °C. It looks that the thermal insulation performance is not good. What's the thermal conductivities of the aerogels. Are the aerogels really thermal insulators? This should be reconsidered.

Reviewer #3 (Remarks to the Author):

The team prepared two types of aerogel (rGO/PPy/PUF and rGO/PPy/PU) and used different hot press techniques (uniaxial, biaxial, and triaxial) to change their microstructure. They ended up with a microstructure that had a zero or negative poisson ratio. They subjected the samples to a standard tensile test and also performed stretching-releasing cycles with a 500% tensile strain. The article includes in-situ morphology images during tensile and compression tests. They also conducted FEA simulations on the samples. At the end part of the article, there are sections discussing the sensing properties, smart thermal management, and smart EMI shielding of the aerogels.

They group has published similar papers. A clear description of the novelty over their previous publications is needed.

The team needs to expand their search for microstructure-tuned aerogels such as meta-aerogels showing high stretch ratios (up to 5400%) and negative Poisson's ratio. Some examples are included herein.

Meta-aerogels: Auxetic shape-memory polyurethane aerogels
S Malakooti, ABMS ud Doulah, Y Ren, VN Kulkarni... - ACS Applied Polymer Materials, 2021

Superelastic graphene aerogel-based metamaterials,
Mingmao Wu, Hongya Geng, Yajie Hu, Hongyun Ma, Ce Yang, Hongwu Chen, Yeye Wen, Huhu Cheng, Chun Li, Feng Liu, Lan Jiang & Liangti Qu, Nature Communications volume 13, Article number: 4561 (2022)

The authors indicated that their aerogel can be stretched to 1200% strain, yet did not include a figure showing relevant results in the main body. A figure needs to be included to support this statement.

Dear reviewers,

We greatly appreciate the constructive comments provided by the reviewers. We acknowledge the improvements that the comments have been made to our manuscript. Our responses to each comment are listed as follows.

Reviewer #1

1. This material is not an “aerogel”. Although you started with a freeze-dried material but at the end you work with a compressed material which is no longer has any aerogel characteristics. Therefore, comparing this material with aerogel materials is not apple-to-apple comparison. I think the material should be classified as an elastomer, not an aerogel.

Response:

Thank you for your comments. The pristine rGO/PPy/PUF and rGO/PPy/PU aerogels without hot pressing were prepared via ambient pressure drying and freeze drying, respectively. As shown in the SEM images (Fig. 2), the foam- or honeycomb-like highly porous structures are converted into folded and reentrant porous structures with smaller pores after uniaxial, biaxial, and triaxial hot pressing. The pore sizes of rGO/PPy/PUF1 and rGO/PPy/PU are mainly in the range of 100-500 μm and 50-200 μm , respectively. By contrast, the pore sizes of rGO/PPy/PUF1-UHP, rGO/PPy/PUF1-BHP, and rGO/PPy/PUF1-THP are mainly in the range of 30-150 μm , 25-150 μm , and 20-100 μm , respectively, while those of rGO/PPy/PU-UHP, rGO/PPy/PU-BHP, and rGO/PPy/PU-THP are mainly in the range of 20-150 μm , 20-130 μm , and 10-80 μm , respectively. Since the network skeletons become denser and the pores become smaller, the resultant hot-pressed materials show relatively higher bulk densities and lower porosities.

All the pristine rGO/polymer composite aerogels show low bulk densities. As shown in Supplementary Fig. 12, the bulk densities of rGO/PPy/PUF1, rGO/PPy/PUF2, rGO/PVA/MF, and rGO/PPy/PU are 90, 40, 19, and 77 mg cm^{-3} , respectively. After hot pressing, the bulk densities of rGO/PPy/PUF1-UHP, rGO/PPy/PUF1-BHP, rGO/PPy/PUF1-THP, rGO/PPy/PUF2-UHP, rGO/PVA/MF-UHP, rGO/PPy/PU-UHP, rGO/PPy/PU-BHP, and rGO/PPy/PU-THP are 370, 370, 700, 280, 70, 330, 330, and 650 mg cm^{-3} , respectively.

The porosities of different materials are calculated by the following equation,

$$P = \left(1 - \frac{\rho_0}{\rho}\right) \times 100\% \quad (1)$$

where P is porosity, ρ_0 is bulk density of the porous materials, and ρ is skeletal density. All of the pristine rGO/polymer composite aerogels show high porosities. The porosities of rGO/PPy/PUF1,

rGO/PPy/PUF2, rGO/PVA/MF, and rGO/PPy/PU are 91%, 96%, 98%, and 92%, respectively. After hot pressing, the porosities of rGO/PPy/PUF1-UHP, rGO/PPy/PUF1-BHP, rGO/PPy/PUF1-THP, rGO/PPy/PUF2-UHP, rGO/PVA/MF-UHP, rGO/PPy/PU-UHP, rGO/PPy/PU-BHP, and rGO/PPy/PU-THP are 63%, 63%, 30%, 72%, 95%, 67%, 67%, and 35%, respectively. The porosities of all the uniaxially and biaxially hot-pressed materials are higher than 60%. In particular, the porosity of the hot-pressed rGO/PVA/MF-UHP is higher than 90%. As we can see, although the bulk density increases and porosity decreases after hot pressing, the resultant materials still possess porous structures. Therefore, in the revised manuscript, we classify the hot-pressed materials as porous materials or porous elastomers.

In order to make a comprehensive comparison, the stretchability of our hot-pressed porous elastomers is compared not only with those of the reported stretchable aerogels but also with those of the reported other stretchable porous materials such as foams and sponges (Fig. 3i,j and Supplementary Table 6).

The above data and relevant discussions have been added to the revised manuscript and supplementary information.

2. The material was not characterized in terms of density, porosity, pore size and surface area, before and after hot pressing process. Just because you freeze-dry your material, it is not guaranteed you end up to a high surface area material. So, these characterizations should be conducted.

Response:

Thank you for your comments. As shown in Supplementary Fig. 12, the bulk densities of rGO/PPy/PUF1, rGO/PPy/PUF2, rGO/PVA/MF, and rGO/PPy/PU are 90, 40, 19, and 77 mg cm⁻³, respectively, confirming the low densities of the pristine rGO/polymer composite aerogels. After hot pressing, the bulk densities of rGO/PPy/PUF1-UHP, rGO/PPy/PUF1-BHP, rGO/PPy/PUF1-THP, rGO/PPy/PUF2-UHP, rGO/PVA/MF-UHP, rGO/PPy/PU-UHP, rGO/PPy/PU-BHP, and rGO/PPy/PU-THP are 370, 370, 700, 280, 70, 330, 330, and 650 mg cm⁻³, respectively.

Supplementary Fig. 12. Bulk densities of typical porous materials. a) Bulk densities of PUF1, rGO/PPy/PUF1, and hot-pressed rGO/PPy/PUF composite materials. b) Bulk densities of PUF2, rGO/PPy/PUF2, and rGO/PPy/PUF2-UHP. c) Bulk densities of MF, rGO/PVA/MF, and rGO/PVA/MF-UHP. d) Bulk densities of rGO/PU, rGO/PPy/PU, and hot-pressed rGO/PPy/PU composite materials.

The porosities of different materials are summarized in Supplementary Fig. 13. The porosities of rGO/PPy/PUF1, rGO/PPy/PUF2, rGO/PVA/MF, and rGO/PPy/PU reach 91%, 96%, 98%, and 92%, respectively. After hot pressing, the porosities of rGO/PPy/PUF1-UHP, rGO/PPy/PUF1-BHP, rGO/PPy/PUF1-THP, rGO/PPy/PUF2-UHP, rGO/PVA/MF-UHP, rGO/PPy/PU-UHP, rGO/PPy/PU-BHP, and rGO/PPy/PU-THP are 63%, 63%, 30%, 72%, 95%, 67%, 67%, and 35%, respectively. The porosities of all the uniaxially and biaxially hot-pressed materials are higher than 60%. In particular, the porosity of the hot-pressed rGO/PVA/MF-UHP is higher than 90%. As we can see, although the bulk density increases and porosity decreases after hot pressing, the resultant materials still possess porous structures.

Supplementary Fig. 13. Porosities of typical porous materials. a) Porosities of PUF1, rGO/PPy/PUF1, and hot-pressed rGO/PPy/PUF composite materials. b) Porosities of PUF2, rGO/PPy/PUF2, and rGO/PPy/PUF2-UHP. c) Porosities of MF, rGO/PVA/MF, and rGO/PVA/MF-UHP. d) Porosities of rGO/PU, rGO/PPy/PU, and hot-pressed rGO/PPy/PU composite materials.

As shown in the SEM images (Fig. 2), the foam- or honeycomb-like highly porous structures are converted into folded and reentrant porous structures with smaller pores after uniaxial, biaxial, and triaxial hot pressing. The pore sizes of rGO/PPy/PUF1 and rGO/PPy/PU are mainly in the range of 100-500 μm and 50-200 μm , respectively. By contrast, the pore sizes of rGO/PPy/PUF1-UHP, rGO/PPy/PUF1-BHP, and rGO/PPy/PUF1-THP are mainly in the range of 30-150 μm , 25-150 μm , and 20-100 μm , respectively, while those of rGO/PPy/PU-UHP, rGO/PPy/PU-BHP, and rGO/PPy/PU-THP are mainly in the range of 20-150 μm , 20-130 μm , and 10-80 μm , respectively.

The nitrogen adsorption-desorption isotherms of typical porous materials before and after hot pressing have been measured. However, the specific surface areas of rGO/PPy/PUF1, rGO/PPy/PUF1-UHP, rGO/PPy/PU, and rGO/PPy/PU-UHP are lower than 1 $\text{m}^2 \text{g}^{-1}$, indicating that there are almost no micropores and mesopores. The nitrogen adsorption-desorption isotherm of rGO/PPy/PUF1 is shown in Fig. I.

Fig. I. Nitrogen adsorption-desorption isotherm of rGO/PPy/PUF1.

The above data and relevant discussions have been added in the revised manuscript and supplementary information.

3. What is the mechanism for shape fixing at high temperature? Is it a reversible shape fixing process if the material is heated back to high temperature? Is it dominated by the PU foam matrix? What the glass transition looks like for the material? It seems there might be multiple transitions.

Response:

Thank you for your comments. Softening and plastic deformations of the polymers (PUF, PU, PVA, or MF) in the skeletons of the rGO/polymer nanocomposites will occur when the porous materials were compressed and heated at 120-140 °C. The polymer chains tend to be deformed and move at high temperatures upon compression, and the shape of the polymer chains after deformation can be fixed when cooling down to room temperature after hot pressing. This will facilitate the shape fixing of the compressed rGO/polymer skeletons after hot pressing. Besides, as shown in the optical microscope and SEM images (Fig. 2 and Supplementary Figs. 9-11), new contacts between the neighboring pore walls of the elastomers can be produced after hot pressing, which will result in further physical crosslinking (such as hydrogen bonds) at these contact points. This may also contribute to the shape fixing of the aerogels after hot pressing. Since the skeletons of the rGO/PPy/PUF and rGO/PPy/PU composite materials are mainly composed of PU, their shape fixing processes are supposed to be dominated by the PU matrix.

To study the transition process during hot pressing, the storage modulus, loss modulus, and tan

delta of rGO/PPy/PUF1 and rGO/PPy/PU under different temperatures were measured by a thermomechanical analyzer (DMA Q800, TA Instruments, USA) (Supplementary Fig. 14). For rGO/PPy/PUF1, the peak of tan delta is located at approximately 110 °C. Accordingly, its storage modulus decreases obviously at around 110 °C, indicating that a softening process occurs at approximately 110 °C. For rGO/PPy/PU, the broad peak of tan delta is located at approximately 70-180 °C. Meanwhile, the storage modulus of rGO/PPy/PU decreases with the increase of temperature in the range of 50-130 °C. These results confirm that the rGO/polymer composite materials will undergo a softening process during hot pressing at 140 or 120 °C.

Supplementary Fig. 14. Storage modulus, loss modulus, and tan delta of a) rGO/PPy/PUF1 and b) rGO/PPy/PU.

As the reviewer commented, the shape fixing process of these materials are reversible. As shown in Supplementary Fig. 15, when rGO/PPy/PUF1-UHP is stretched 200% and heated at 140 °C for 5 h under vacuum condition, its shape can be fixed without springing back after cooling down to room temperature. The resultant material can be hot pressed (140 °C for 5 h under vacuum condition) to its initial size of rGO/PPy/PUF1-UHP. This confirms the reversible shape fixing of the rGO/polymer composite materials.

Supplementary Fig. 15. Reversible shape fixing process of rGO/PPy/PUF1-UHP.

The above data and discussions have been added to the revised manuscript and supplementary information.

4. In figure 1, a chemical structure was shown. However, no data is provided to support the structure. How do you know if this is what you have instead of just an intertwined multi-strand network?

Response:

Thank you for your comments. More data have been provided to support the structure in Fig. 1. The crosslinks between GO and APTES via a C-N coupling reaction and hydrolytic polycondensation of alkoxy groups have been revealed in our previous work.^[62] To further study the structure of the rGO/polymer aerogels, X-ray photoelectron spectra (XPS) of the APTES-crosslinked rGO/PU and rGO aerogels as well as GO/PU aerogel without APTES were measured via an X-ray photoelectron spectrometer (ESCALAB Xi+, Thermo Scientific, USA) (Supplementary Figs. 17-19 and Table 4).

Supplementary Fig. 17. a) XPS spectrum of the APTES-crosslinked rGO/PU aerogel. b) XPS O 1s, c) N 1s, d) C 1s, and e) Si 2p spectra of the APTES-crosslinked rGO/PU aerogel.

Supplementary Fig. 18. a) XPS spectrum of the APTES-crosslinked rGO aerogel. b) XPS O 1s, c) N 1s, d) C 1s, and e) Si 2p spectra of the APTES-crosslinked rGO aerogel.

Supplementary Fig. 19. a) XPS spectrum of the GO/PU aerogel without APTES. b) XPS O 1s, c) N 1s, and d) C 1s spectra of the GO/PU aerogel without APTES.

XPS spectrum of the APTES-crosslinked rGO/PU and rGO aerogels clearly shows five peaks at 532.1, 399.3, 284.3, 152.8 and 101.1 eV, corresponding to O 1s, N 1s, C 1s, Si 2s, and Si 2p, respectively (Supplementary Figs. 17, 18).^[59] For the GO/PU aerogel without APTES, there are three peaks at 532.0, 398.9, and 284.3 eV, corresponding to O 1s, N 1s, and C 1s, respectively (Supplementary Fig. 19), where the peak of N 1s is attributed to PU. There are no peaks corresponding to Si 2s and Si 2p in the GO/PU aerogel without APTES (Supplementary Fig. 19). The resultant percentage of element content of different aerogels is listed in Supplementary Table 4.

Supplementary Table 4. Percentage of element content by XPS surveys of different kinds of aerogels.

Sample	C 1s (%)	O 1s (%)	N 1s (%)	Si 2p (%)
APTES-crosslinked rGO/PU aerogel	72.44	21.04	4.25	2.28
APTES-crosslinked rGO aerogel	56.22	26.57	8.40	8.81

GO/PU aerogel without APTES	74.92	23.37	1.71	0
-------	-------	------	---

The O 1s spectrum of the APTES-crosslinked rGO/PU aerogel can be divided into three peaks at 532.7, 532.1, and 531.4 eV, representing C-O, Si-O, and C=O bonds, respectively (Supplementary Fig. 17).^[60] The N 1s spectra of the APTES-crosslinked rGO/PU aerogel and GO/PU aerogel without APTES show only one peak at 398.9 eV, representing -NH- bond.^[61] The N 1s spectrum of the APTES-crosslinked rGO aerogel shows two peaks at 401.2 and 398.9 eV, representing -NH₂/NH₃⁺ and -NH- bonds, respectively.^[61] The -NH- bond of the APTES-crosslinked rGO aerogel is supposed to be attributed to the nucleophilic displacement reaction between epoxy groups in GO and amino groups in APTES (Fig. 1b and Supplementary Fig. 2) ^[61,62]. The -NH- bond of the APTES-crosslinked rGO/PU aerogel is supposed to come from the -COO-NH- groups of PU and the nucleophilic displacement reaction between epoxy groups in GO and amino groups in APTES. The C 1s spectra of the APTES-crosslinked rGO/PU aerogel can be divided into five peaks at 288.0, 286.1, 285.4, 284.1, and 283.5 eV, representing C=O, C-O, C-N, C-C, and C-Si bonds, respectively (Supplementary Fig. 17).^[17,61,63] For the APTES-crosslinked rGO/PU and rGO aerogels, the Si 2p spectrum can be divided into two peaks at 101.1 and 101.7 eV, corresponding to Si-C and Si-O-Si bonds, respectively (Supplementary Figs. 17, 18).^[64] The Si-O-Si bond is attributed to the hydrolytic polycondensation of APTES. These results indicate that GO nanosheets can be crosslinked by APTES via a C-N coupling reaction between epoxy groups of GO and amino groups of APTES combined with the hydrolytic polycondensation of alkoxy groups of APTES. Besides, the gel forms more quickly after adding APTES in the GO dispersion, providing an indirect evidence of the crosslinks between APTES and GO nanosheets.

In addition, as shown in the FTIR spectrum of the rGO/PU aerogel (Supplementary Fig. 12h), the peaks at 1533 and 1069 cm⁻¹ correspond to the N-H and C-O-C groups, respectively. ^[58]

It is found that the rGO/PU gel forms more quickly compared with rGO gel without PU, indicating that there may be physical crosslinks between rGO and PU, which can be achieved by the hydrogen bonds between -NH-COO- groups in PU and -COOH (or -OH) groups in GO.

The PPy particles remain attached to the skeleton of rGO/PU even after repeated washing, squeezing, hot pressing, stretching, and compression. Additionally, the conductivity of

rGO/PPy/PU-UHP remains nearly unchanged after repeated stretching and compression. This indicates that there may be physical crosslinks between PPy and rGO (or PU), which can be achieved by the hydrogen bonds between -NH groups in PPy and -COOH (or -OH) groups in GO (or -NH-COO- groups in PU).

Supplementary References

[58] Yu, Y., Zhai, Y., Yun, Z., Zhai, W., Wang, X., Zheng, G., Yan, C., Dai, K., Liu, C., Shen, C. Ultra-stretchable porous fiber-shaped strain sensor with exponential response in full sensing range and excellent anti-interference ability toward buckling, torsion, temperature, and humidity. *Adv. Electron. Mater.* 5, 1900538 (2019).

[59] Wu, X., Fan, M., Shen, X., Cui, S. & Tan, G. Silica aerogels formed from soluble silicates and methyl trimethoxysilane (MTMS) using CO₂ gas as a gelation agent. *Ceram. Int.* 44, 821-829 (2018).

[60] Smith, M., Scudiero, L., Espinal, J. & McEwen, J. Improving the deconvolution and interpretation of XPS spectra from chars by ab initio calculations. *Carbon* 110, 155-171 (2016).

[61] Hu, H., Zhao, Z., Wan, W., Gogotsi, Y. & Qiu, J. Ultralight and highly compressible graphene aerogels. *Adv. Mater.* 25, 2219-2223 (2013).

[62] Zu, G., Kanamori, K., Nakanishi, K., Lu, X., Yu, K., Huang, J. & Sugimura, H. Superelastic multifunctional aminosilane-crosslinked graphene aerogels for high thermal insulation, three-component separation, and strain/pressure-sensing arrays. *ACS Appl. Mater. Interfaces* 11, 43533-43542 (2019).

[63] Bashouti, M. Y, Paska, Y., Puniredd, S. R., Thomas, T., Christiansen, S. & Haick, H. Silicon nanowires terminated with methyl functionalities exhibit stronger Si-C bonds than equivalent 2D surfaces. *Phys. Chem. Chem. Phys.* 11, 3845-3848 (2009).

[64] Gupta, P., Sathwane, M., Chhajed, M., Verma, C., Grohens, Y., Seantier, B., Agrawal, A. K & Maji, P. K. Surfactant assisted in situ synthesis of nanofibrillated cellulose/polymethylsilsesquioxane aerogel for tuning its thermal performance. *Macromol. Rapid Commun.* 44, 2200628 (2023).

The above data and discussions have been added to the revised manuscript and supplementary information.

Reviewer #2

1. As a composite aerogel, all the compositions and the interfaces between them in the aerogel are responsible for the mechanical property and functions. The current results in the characterizations of the structures of the aerogels are not sufficient. For examples, the interface between the polymer and rGO is not clearly, the morphology and crystal size of the rGO are not mentioned. These structure characters are very important for the mechanical and functional properties.

Response:

Thank you for your comments. To further characterize the structures of the aerogels, X-ray photoelectron spectra (XPS) of the APTES-crosslinked rGO/PU and rGO aerogels as well as GO/PU aerogel without APTES were measured (Supplementary Figs. 17-19 and Table 4).

XPS spectrum of the APTES-crosslinked rGO/PU and rGO aerogels clearly shows five peaks at 532.1, 399.3, 284.3, 152.8 and 101.1 eV, corresponding to O 1s, N 1s, C 1s, Si 2s, and Si 2p, respectively (Supplementary Figs. 17, 18).^[59] For the GO/PU aerogel without APTES, there are three peaks at 532.0, 398.9, and 284.3 eV, corresponding to O 1s, N 1s, and C 1s, respectively (Supplementary Fig. 19), where the peak of N 1s is attributed to PU. There are no peaks corresponding to Si 2s and Si 2p in the GO/PU aerogel without APTES (Supplementary Fig. 19).

The O 1s spectrum of the APTES-crosslinked rGO/PU aerogel can be divided into three peaks at 532.7, 532.1, and 531.4 eV, representing C-O, Si-O, and C=O bonds, respectively (Supplementary Fig. 17).^[60] The N 1s spectra of the APTES-crosslinked rGO/PU aerogel and GO/PU aerogel without APTES show only one peak at 398.9 eV, representing -NH- bond.^[61] The N 1s spectrum of the APTES-crosslinked rGO aerogel shows two peaks at 401.2 and 398.9 eV, representing -NH₂/NH₃⁺ and -NH- bonds, respectively.^[61] The -NH- bond of the APTES-crosslinked rGO aerogel is supposed to be attributed to the nucleophilic displacement reaction between epoxy groups in GO and amino groups in APTES (Fig. 1b and Supplementary Fig. 2) ^[61,62]. The -NH- bond of the APTES-crosslinked rGO/PU aerogel is supposed to come from the -COO-NH- groups of PU and the nucleophilic displacement reaction between epoxy groups in GO and amino groups in APTES. The C 1s spectra of the APTES-crosslinked rGO/PU aerogel can be divided into five peaks at 288.0, 286.1, 285.4, 284.1, and 283.5 eV, representing C=O, C-O, C-N, C-C, and C-Si bonds, respectively (Supplementary Fig. 17).^[17,61,63] For the APTES-crosslinked rGO/PU and rGO aerogels, the Si 2p spectrum can be divided into two peaks at 101.1 and 101.7 eV, corresponding to Si-C and Si-O-Si bonds, respectively (Supplementary Fig. 17, 18).^[64] The Si-O-Si bond is attributed to the hydrolytic polycondensation of APTES. These results indicate that GO nanosheets can be crosslinked by

APTES via a C-N coupling reaction between epoxy groups of GO and amino groups of APTES combined with the hydrolytic polycondensation of alkoxy groups of APTES.

As shown in SEM and optical microscope images of pristine rGO/PPy/PUF aerogels and hot-pressed porous rGO/PPy/PUF elastomers (Supplementary Figs. 9, 10), the crosslinked rGO nanosheets are well deposited on the surface of the PUF skeletons and some pore walls of PUF are connected by the rGO nanosheets. rGO nanosheets and PUF skeletons are well connected with each other at the interfaces between them. The size of the rGO nanosheets in rGO/PPy/PUF system is mainly in the range of 50-500 μm , while that of the rGO nanosheets in rGO/PVA/MF system is mainly in the range of 20-200 μm (Supplementary Fig. 9). In the case of rGO/PU system, rGO and PU are well combined and no obvious interfaces are observed in SEM images (Supplementary Fig. 9). It is found that the rGO/PU gel forms more quickly compared with rGO gel without PU, indicating that there may be physical crosslinks between rGO and PU, which can be achieved by the hydrogen bonds between -NH-COO- groups in PU and -COOH (or -OH) groups in GO.

There are many irregular spherical particles on the surface of the skeletons of the aerogels rGO/PPy/PUF1, rGO/PPy/PUF2, and rGO/PPy/PU, which are well preserved after hot pressing (Fig. 2i-p and Supplementary Fig. 10). These particles are supposed to be PPy polymers, which were deposited by in-situ oxidation polymerization of pyrrole. The interface between PPy particles and rGO (or PU) is clearly observed in SEM images of the samples (Fig. 2 and Supplementary Fig. 10). The PPy particles remain attached to the skeleton of rGO/PU even after repeated washing, squeezing, hot pressing, stretching, and compression. Additionally, the conductivity of rGO/PPy/PU-UHP remains nearly unchanged after repeated stretching and compression. This indicates that there may be physical crosslinks between PPy and rGO (or PU), which can be achieved by the hydrogen bonds between -NH- groups in PPy and -COOH (or -OH) groups in GO (or -NH-COO- groups in PU).

The above data and discussions have been added to the revised manuscript and supplementary information.

2. Densities and porosities of the composite aerogels are missed. These parameters are very important for the determinations and evaluation of their mechanical properties and thermal insulation performance.

Response:

Thank you for your comments. As shown in Supplementary Fig. 12, the bulk densities of rGO/PPy/PUF1, rGO/PPy/PUF2, rGO/PVA/MF, and rGO/PPy/PU are 90, 40, 19, and 77 mg cm^{-3} ,

respectively, confirming the low bulk densities of the pristine rGO/polymer composite aerogels. After hot pressing, the bulk densities of rGO/PPy/PUF1-UHP, rGO/PPy/PUF1-BHP, rGO/PPy/PUF1-THP, rGO/PPy/PUF2-UHP, rGO/PVA/MF-UHP, rGO/PPy/PU-UHP, rGO/PPy/PU-BHP, and rGO/PPy/PU-THP are 370, 370, 700, 280, 70, 330, 330, and 650 mg cm⁻³, respectively. The resultant hot-pressed materials show relatively higher bulk densities.

The porosities of rGO/PPy/PUF1, rGO/PPy/PUF2, rGO/PVA/MF, and rGO/PPy/PU reach 91%, 96%, 98%, and 92%, respectively (Supplementary Fig. 13). After hot-pressing, the porosities of rGO/PPy/PUF1-UHP, rGO/PPy/PUF1-BHP, rGO/PPy/PUF1-THP, rGO/PPy/PUF2-UHP, rGO/PVA/MF-UHP, rGO/PPy/PU-UHP, rGO/PPy/PU-BHP, and rGO/PPy/PU-THP are 63%, 63%, 30%, 72%, 95%, 67%, 67%, and 35%, respectively. The porosities of all the uniaxially and biaxially hot-pressed materials are higher than 60%. In particular, the porosity of the hot-pressed rGO/PVA/MF-UHP is higher than 90%. As we can see, although the bulk density increases and porosity decreases after hot pressing, the resultant materials still possess porous structures.

In addition, the comments on the influence of bulk density and porosity on their mechanical properties and thermal insulation performance have been added to the revised manuscript and supplementary information. These comments are shown below.

“The compressed skeletons with smaller pores, higher bulk densities, and lower porosities contribute to the higher compressive strength of the hot-pressed elastomers.”

“The larger pore size and lower apparent density under a larger tensile strain may result in the higher thermal conductivities of gas and radiation, leading to the lower thermal insulation of the hot-pressed aerogel with larger tensile strains (Fig. 6c).”

“The apparent density of the hot-pressed rGO/polymer aerogel decreased with increasing tensile strain, which would result in lower λ_s The pore size and porosity of the hot-pressed rGO/polymer aerogel increased with increasing tensile strain, which would lead to higher λ_g The decreased apparent density of the hot-pressed aerogel with a larger tensile strain would result in higher λ_r . The increased λ_g and λ_r probably contributed to the lower thermal insulation of the hot-pressed rGO/polymer aerogel with larger tensile strains.”

3. As the authors claim that “Softening and plastic deformations of the polymers (PUF, PU, PVA, or MF) in the skeletons of the rGO/polymer aerogels would occur when the aerogels were compressed and heated at 120-140 °C”, strains in the structure might be released during the softening process. Therefore, simply using FEA results to reflect the strain distribution (Figure 4c) in the microstructure of the aerogel is not accurate.

Response:

Thank you for your comments. As the reviewer commented, strains in the structure might be released during the softening process. The FEA mechanical simulations were performed based on simplified hot-pressing and stretching processes without considering the possible strain releasing during the softening process. The actual strain distribution after hot pressing may be different from the simulation results when considering the softening process during hot pressing. In the case of hot-pressing simulations, the pores of the elastomers are compressed and the pore walls are folded gradually upon uniaxial hot pressing (Fig. 4c and Supplementary Movie 8). Reentrant porous structures of the elastomers are formed after biaxial hot pressing (Fig. 4c and Supplementary Movie 9). The simulated folded and reentrant structures after uniaxial and biaxial hot pressing are generally consistent with the morphologies of the corresponding hot-pressed rGO/polymer elastomers as presented in Fig. 2 and Supplementary Figs. 9-11. As we can see, the FEA results can roughly reflect the microstructures after uniaxial and biaxial hot pressing.

The above discussion has been added to the revised manuscript.

4. To explain the reasons for the high stretchability, the authors showed three main reasons. “First, the flexibility of the polymers (PUF, PU, PVA, or MF) endows the rGO/polymer networks with excellent reversible deformability, which can allow the aerogels to be compressed with a large strain without fracture during hot pressing. Second, the compressed rGO/polymer networks can be fixed without springing back after being cooled to room temperature. Third, the obtained folded or reentrant porous rGO/polymer structures after hot pressing are highly stretchable.” One can find that this kind of explanation is used phenomena to explain phenomena. The stretchability is the mechanical property of the composite aerogel. The author should analyze the deformation process from observe the microstructure characters of the aerogels themselves and microstructure changes during tensile deformation.

Response:

Thank you for your comments. We have analyzed the deformation process by observing the microstructure characters of the porous materials and microstructure changes during tensile deformation to explain the reasons for the high stretchability, which is shown below.

“As presented in the in-situ optical microscope images of the hot-pressed rGO/polymer elastomers during stretching (Fig. 4a, b and Supplementary Fig. 37), the tensile deformation of the hot-pressed porous elastomers can be divided into three processes. The first deformation process is in the strain range of approximately 0-5%, during which many connected neighboring pore walls will be disconnected by overcoming the intramolecular or intermolecular interactions (such as hydrogen bonds). In this process, the stress increases significantly with the increase of tensile strain (Figure

3d-h). The second deformation process is in the strain range of approximately 5% to $(1/(1-\varepsilon_0)-1)\times 100\%$, where ε_0 is the compressive strain during hot pressing. For rGO/PPy/PUF1-UHP, $\varepsilon_0=75\%$ and $(1/(1-\varepsilon_0)-1)\times 100\%=300\%$. In this process, the folded and reentrant pores become unfolded and are gradually stretched to nearly their original shapes before hot pressing. This unfolding process can undergo large tensile strains without fracture. The tensile strain during the third deformation process is approximately larger than $(1/(1-\varepsilon_0)-1)\times 100\%$. In this process, the pores are further stretched along the direction of tensile strain. The deformability in this process is determined by the stretchability of pristine rGO/polymer aerogels without hot pressing. The high deformability of the folded and reentrant microstructures during the second and third deformation processes mainly contribute to the high stretchability of the hot-pressed porous rGO/polymer elastomers. The high deformability of the polymers (such as PUF, PU, and MF) and the good combination of the polymers and rGO also contribute to their high stretchability.”

The above discussion has been added to the revised manuscript.

5. What does the “intrinsically stretchability” mean? For most inorganic materials, large strain deformation may not their intrinsic properties. The stretchability of most graphene based aerogels, and nanofibers based aerogels are mostly from their nanometer scale building units. Moreover, the deformation ability of polymeric and inorganic materials is different, their mechanical properties can not be simply comparee.

Response:

Thank you for your comments. There is a mistake for us to call the stretchability of our porous materials “intrinsic stretchability” in our initial manuscript. In this work, the stretchability of the hot-pressed porous elastomers is supposed to be contributed mainly by the microstructural building units. As the reviewer commented, for most inorganic materials, large deformation may not their intrinsic properties. Therefore, “intrinsic” and “intrinsically” have been removed from the revised manuscript. “intrinsic stretchability” and “intrinsically stretchable” have been changed to “stretchability” and “stretchable”, respectively.

In this work, in order to make a comprehensive comparison, the stretchability of our hot-pressed organic-inorganic porous elastomers are compared not only with those of polymeric and organic-inorganic porous materials but also with those of inorganic porous materials.

6. A high accuracy is very important for a strain or pressure sensor. The demands for strain sensor and stress sensor might be different. A material that is suitable for strain sensor may not be suitable for stress sensor. During the 1000 stretching-releasing cycles, it could be seen that the stress decrease

significantly. One could question that would the present aerogel be suitable for sensors. Moreover, for a very large strain, like 500% or 1000% tensile strain, would it need a sensor to detect? Is there any real application situations?

Response:

Thank you for your comments. As the reviewer commented, for rGO/PPy/PUF1-UHP, the stress at 500% strain indeed decreases from 121 kPa to 59 kPa after stretching-releasing with 500% tensile strain for 1000 cycles (Fig. 3f). However, rGO/PPy/PUF1-UHP is highly elastic and nearly recovers its original shape after stretching-releasing in x direction with 500% strain for 1000 cycles. The response of the rGO/PPy/PUF1-UHP-based strain sensor remained nearly unchanged during stretching-releasing in x direction with 400% strain for 1000 cycles, indicating its high durability and fatigue resistance (Fig. 5d). This makes them suitable for broad-range-response strain sensors for detecting tensile strains.

While the strain is loaded along x direction for the strain sensor, the pressure is loaded along z direction for the pressure sensor. As shown in Supplementary Fig. 42, the response of rGO/PPy/PUF1-UHP-based pressure sensor remains nearly unchanged after compressing-releasing in z direction with 80% strain for 1000 cycles. However, the compressive stress still decreased after repeated compression-decompression (Supplementary Fig. 32). The stability of the pressure sensors needs to be further enhanced. The measured lower detection limit of the rGO/PPy/PUF1-UHP-based pressure sensor is 1.5 kPa. Therefore, the rGO/PPy/PUF1-UHP-based pressure sensor shows a broad detection range of 1.5 kPa-4.7 MPa.

Strain sensors with a large detection range can be used for monitoring large tensile strains of elastic rope during bungee jumping, chest developer during exercise, coil spring during stretching, soft robot, etc. The tensile strain of a chest developer during exercise can reach at least 300% (Fig. 5h). The coil spring in some mechanical equipment during stretching or compression can reach 500-1000%. Another example is that advanced stretchable soft robots may undergo large tensile strains during working.

The above discussions have been added to the revised manuscript.

7. For the thermal insulation performance of the aerogel, it could be seen that the surface of the aerogel reached 42.4 °C in 30s when the substrate is only 50 °C. It looks that the thermal insulation performance is not good. What's the thermal conductivities of the aerogels. Are the aerogels really thermal insulators? This should be reconsidered.

Response:

Thank you for your comments. For the thermal insulation demonstration in Fig. 6 in the pristine manuscript, the thickness of the used rGO/PPy/PUF1-UHP is only 1 mm, resulting in the relative high temperature (42.4 °C) of the top surface of the sample in 30 s. The real-time infrared thermal images and temperatures of 2 mm thick rGO/PPy/PUF1-UHP with different tensile strains on a hot plate (51 °C) are added in Fig. 6a,b in the revised manuscript. When the thickness of the sample is 2 mm, the top surface temperature of pristine rGO/PPy/PUF1-UHP stabilizes at 36.1 °C after 60 s, the value of which is lower than that of the 1 mm thick sample (42.4 °C in 30 s), indicating a better thermal insulation performance. When the sample is stretched 250%, the top surface temperature stabilizes at 42.0 °C, the value of which is higher than those of the pristine sample and stretched sample with tensile strains of 80%, 150%, and 180%.

Fig. 6a. Real-time infrared thermal images of pristine and stretched rGO/PPy/PUF1-UHP (2 mm thick) with different tensile strains on a hot plate.

Fig. 6b. Real-time temperatures of pristine and stretched rGO/PPy/PUF1-UHP (2 mm thick) with different tensile strains on a hot plate.

In addition, the thermal conductivities were measured by the plate thermal flow method using a thermal conductivity tester (DRPL-III, Xiangtan Xiangyi instrument Co., Ltd, P. R. China). The thermal conductivities of rGO/PPy/PUF1-UHP and rGO/PPy/PU-UHP at room temperature and ambient pressure are 0.034 and 0.029 W m⁻¹ K⁻¹, respectively, the values of which are lower than those of some commercial thermal insulation materials such as mineral wool and comparable to those of extruded polystyrene and polyurethane foam, indicating that they are thermal insulators.^[5]

Reviewer #3

1. They group has published similar papers. A clear description of the novelty over their previous publications is needed.

Response:

Thank you for your comments. Our group has published a paper entitled “Bioinspired gradient stretchable aerogels for ultrabroad-range-response pressure-sensitive wearable electronics and high-efficient separators” (10.1002/anie.202213952) (Ref. [22]). In our previous work,^[22] uniaxial hot pressing is used to enhance the modulus of the high-modulus layer of the gradient rGO/PUF composite aerogels instead of their stretchability. In this work, uniaxial, biaxial, and triaxial hot-pressing strategies are used to enhance stretchability and achieve near-zero and negative Poisson’s ratios of the porous materials. The sensors in this work are stretched in the opposite direction of hot pressing, while the stretching direction of the sensors in our previous work is perpendicular to the direction of hot pressing. Besides, in this work, electrical conductivities of the rGO/PUF composite materials are enhanced by the deposition of PPy in the skeletons of PUF or PU.

The above description of the novelty over our previous work is added to the revised manuscript.

2. The team needs to expand their search for microstructure-tuned aerogels such as meta-aerogels showing high stretch ratios (up to 5400%) and negative Poisson's ratio. Some examples are included herein.

Meta-aerogels: Auxetic shape-memory polyurethane aerogels

S Malakooti, ABMS udDoulah, Y Ren, VN Kulkarni... - ACS Applied Polymer Materials, 2021

Superelastic graphene aerogel-based metamaterials,

Mingmao Wu, Hongya Geng, Yajie Hu, Hongyun Ma, Ce Yang, Hongwu Chen, Yeye Wen, Huhu Cheng, Chun Li, Feng Liu, Lan Jiang & Liangti Qu, Nature Communications volume 13, Article number: 4561 (2022).

Response:

Thank you for your comments. The publication entitled “Superelastic graphene aerogel-based metamaterials” (Wu, M., et al. Nat. Commun. 13, 4561 (2022)) has already been cited (Ref. [8]) in the Introduction section of our pristine manuscript. The discussion on their stretchability and negative Poisson’s ratios during compression has been added to the revised manuscript and shown below. “While the graphene aerogels with serpentine and spiral structures show high stretchability up to 1200% and 5400%, respectively, the same graphene aerogels without the special macroscopic structures only exhibit an elongation at break of 6%.^[8]...Graphene aerogels with Poisson’s ratios in the range of -0.95-1.64 during compression can be obtained by constructing meta-structures via laser engraving. ^[8]”

The recommended publication entitled “Meta-aerogels: Auxetic shape-memory polyurethane aerogels” has been cited (Ref. [52]) in the revised manuscript.

[52] Malakooti, S., Doulah, A. B. M. S. U., Ren, Y., Kulkarni, V. N., Soni, R. U., Edlabadkar, V. A., Zhang, R., Vivod, S. L., Sotiriou-Leventis, C., Leventis, N. & Lu, H. Meta-aerogels: Auxetic shape-memory polyurethane aerogels, ACS Appl. Polym. Mater. 3, 5727 (2021).

Additionally, in our revised manuscript, we have cited another publication on microstructure-tuned metamaterials, which is shown below.

[53] Yasuda, H. & Yang, J. Reentrant origami-based metamaterials with negative Poisson’s ratio and bistability. Phys. Rev. Lett. 114, 185502 (2015).

These two articles are introduced as follows: “Negative Poisson’s ratios of metamaterials can be achieved via many methods.^[8,29,32-36,52,53] For example, shape-memory PU meta-aerogels with negative Poisson’s ratios can be obtained by using a suitable mold with a metastructure.^[52] Metamaterials with negative Poisson’s ratios can also be achieved by constructing reentrant 3D

origami structures.^[53] Compared with other methods, hot pressing is a relatively simple method to achieve negative Poisson's ratios. Metamaterials with negative Poisson's ratios show potential applications in soft robot, wave propagation manipulation, minimally invasive medical devices, etc.^[52,53]”

3. The authors indicated that their aerogel can be stretched to 1200% strain, yet did not include a figure showing relevant results in the main body. A figure needs to be included to support this statement.

Response:

Thank you for your comments. The elongation at break of rGO/PPy/PUF2-UHP reaches 1250%. The photographs of rGO/PPy/PUF2-UHP during stretching with 1200% strain is shown in Fig. 3a in the revised manuscript. The stress-strain curve of rGO/PPy/PUF2-UHP during stretching with 1250% strain is shown in Fig. 3d in the revised manuscript. Besides, a movie (Supplementary Movie 2) of a tensile test in x direction on rGO/PPy/PUF2-UHP with 1250% strain is provided in the revised supporting information.

Best regards,

Prof. Guoqing Zu

Department of Polymeric Materials, School of Materials Science and Engineering, Tongji University,
Shanghai, P. R. China

Reviewers' comments:

Reviewer #1 (Remarks to the Author):

the revised manuscript is very well presented. I recommend accepting for publication.

Reviewer #2 (Remarks to the Author):

The authors have done some additional work to improve the manuscript, but I still have some comments on the authors' response.

1. In their response, they wrote that "In the case of rGO/PU system, rGO and PU are well combined and no obvious interfaces are observed in SEM images (Supplementary Fig. 9)". As we know, two objects in contact must undoubtedly have an interface.

2. For rGO, what's the thickness of a piece of the material, what's the size of the grain? SEM and optical images can not reflect this. I suggest the author use STEM or TEM to investigate these microstructures and the interface between rGO and PU, these informations very important in understanding both the mechanical and thermal properties.

3. Since the authors also think that "The actual strain distribution after hot pressing may be different from the simulation results when considering the softening process during hot pressing". Why they display the results? Figure 3c may be meaningless.

4. As the author suggested an application of thermal insulation and management. Because the softening of the materials would take place at high temperatures, will this influence the mechanical properties and thermal stability of the materials in the applied conditions? More work should be done to illustrate this concern.

5. As other reviewer mentioned that the group has published similar work, I also checked the work that the authors mentioned in the response letter, I doubt that if the novelty of the present work could meet up with the demand of Nature Communications.

Dear reviewers,

We greatly appreciate the constructive comments provided by the reviewer #2. We acknowledge the improvements that the comments have been made to our manuscript. Our responses to each comment are listed as follows.

Reviewer #2

1. In their response, they wrote that “In the case of rGO/PU system, rGO and PU are well combined and no obvious interfaces are observed in SEM images (Supplementary Fig. 9)”. As we known, two objects in contact must undoubtedly have an interface.

Response:

Thank you for your comments. As you mentioned, the expression of the sentence “In the case of rGO/PU system, rGO and PU are well combined and no obvious interfaces are observed in SEM images (Supplementary Fig. 9)” is not accurate. It has been changed to “In the case of rGO/PU system, the SEM images indicate the well combination of rGO and PU (Supplementary Fig. 9)”. The interface between rGO and polymer was investigated by TEM images (Supplementary Fig. 22), which have been shown in the revised manuscript (Page 8) and supplementary information (Supplementary Figs 21, 22 and Supplementary Note 4).

Supplementary Fig. 22. TEM images of rGO/PPy/PU-UHP. **a** rGO nanosheets. **b** Interface between rGO and polymer.

2. For rGO, what's the thickness of a piece of the material, what's the size of the grain? SEM and optical images can not reflect this. I suggest the author use STEM or TEM to investigate these microstructures and the interface between rGO and PU, these informations very important in understanding both the mechanical and thermal properties.

Response:

Thank you for your comments. As shown in Supplementary Fig. 21, the morphology of a piece of GO in the GO dispersion was characterized by a transmission electron microscope (TEM, Tecnai F20, FEI, USA). The size of GO nanosheets is in the range of approximately 1 to 6 μm . In addition, the microstructures of rGO and the interface between rGO and polymer were investigated by TEM images. As shown in the TEM images of rGO/PPy/PU-UHP (Supplementary Fig. 22), the thicknesses of the rGO nanosheets are in the range of approximately 55-250 nm, while their sizes are in the range of approximately 1-7 μm . The interface between rGO and polymer can be observed clearly by the TEM images (Supplementary Fig. 22b). As we can see, the polymer and rGO in rGO/PPy/PU-UHP are well combined at the interface.

Supplementary Fig. 21. TEM images of GO sheets in the GO dispersion.

Supplementary Fig. 22. TEM images of rGO/PPy/PU-UHP. **a** rGO nanosheets. **b** Interface between rGO and polymer.

The above data and discussion have been added to the revised manuscript (Page 8) and supplementary information (Supplementary Figs 21, 22 and Supplementary Note 4).

3. Since the authors also thinks that “The actual strain distribution after hot pressing may be different from the simulation results when considering the softening possess during hot pressing”. Why they display the results? Figure 3c may be meaningless.

Response:

Thank you for your comments. To investigate the microstructure variation of the materials during hot pressing, the microstructures of the rGO/polymer composites after uniaxial compression with 75% strain at room temperature (rGO/PPy/PUF1-UP) and the rGO/polymer composites after uniaxial hot pressing with 75% strain at 140 °C (rGO/PPy/PUF1-UHP) were observed by an optical microscope (Supplementary Fig. 16). It is found that there is no obvious difference in microstructures between rGO/PPy/PUF1-UP and rGO/PPy/PUF1-UHP, indicating that the strain release caused by softening process during hot pressing is not obvious and can be negligible. Therefore, the FEA mechanical simulations (Figure 4c, d) can theoretically illustrate the microstructure variation during hot pressing and stretching. The simulated folded and reentrant structures after uniaxial and biaxial hot pressing are generally consistent with the morphologies of the corresponding hot-pressed rGO/polymer elastomers

as presented in Fig. 2 and Supplementary Figs. 9-11. These results indicate that Figure 4c and Figure 4d are meaningful. (The figure that reviewer #2 mentioned here is Figure 4c instead of Figure 3c because there is no FEA simulation in Figure 3c).

Supplementary Fig. 16. Optical microscope images of a) rGO/PPy/PUF1-UP and b) rGO/PPy/PUF1-UHP.

The above data and discussion have been added to the revised manuscript (Pages 8, 13) and supplementary information (Supplementary Fig. 16).

4. As the author suggested an application of thermal insulation and management. Because the soften of the materials would take place at high temperatures, will this influence the mechanical properties and thermal stability of the materials in the applied conditions? More work should be done to illustrate this concern.

Response:

Thank you for your comments. To investigate the thermal stability of rGO/PPy/PUF1-UHP and rGO/PPy/PU-UHP, the thermogravimetric (TG) and derivative thermogravimetric (DTG) curves were measured in air by a TG analyzer (TG 209F3, NETZSCH, Germany) from 30 to 800 °C with a heating rate of 10 K min⁻¹ (Supplementary Fig. 47). rGO/PPy/PUF1-UHP shows a significant weight loss of approximately 75% in the temperature range of 200-400 °C, which is accompanied with an intense

peak of DTG at 297.7 °C. In the case of rGO/PPy/PU-UHP, it shows a significant weight loss of approximately 76% in the temperature range of 300-400 °C, which is accompanied with an intense peak of DTG at 358.3 °C. The TG and DTG curves indicate that rGO/PPy/PUF1-UHP and rGO/PPy/PU-UHP are thermally stable up to approximately 200 and 300 °C, respectively.

Supplementary Fig. 47. TG and DTG curves of a) rGO/PPy/PUF1-UHP and b) rGO/PPy/PU-UHP.

To investigate the influence of heat treatment on mechanical properties of the hot-pressed elastomers, compression-decompression curves of rGO/PPy/PUF1-UHP and rGO/PPy/PU-UHP after heat treatment at different temperatures (50, 100, and 140 °C for 2 h) were tested (Supplementary Fig. 48). After heat treatment at 140 °C, the length of rGO/PPy/PUF1-UHP and rGO/PPy/PU-UHP increases 25% and 10%, respectively, along x direction. By contrast, after heat treatment at 100 °C, the length of rGO/PPy/PUF1-UHP and rGO/PPy/PU-UHP only shows a slight increase along x direction (5% and 2%, respectively). There is no change in size for both rGO/PPy/PUF1-UHP and rGO/PPy/PU-UHP after heat treatment at 50 °C. The compressive stresses at 80% strain for rGO/PPy/PUF1-UHP and rGO/PPy/PU-UHP decrease after heat treatment at 140 °C because of the volume expansion, while their compressive stresses at 80% strain show no obvious change after heat treatment at 100 and 50 °C (Supplementary Fig. 48). Both rGO/PPy/PUF1-UHP and rGO/PPy/PU-UHP after heat treatment at different temperatures (50, 100, and 140 °C) maintain high elasticities (Supplementary Fig. 48).

Supplementary Fig. 48. Stress-strain curves of typical hot-pressed elastomers with 80% compressive strain in z direction. a, c, e) Stress-strain curves of rGO/PPy/PUF1-UHP after heat treatment at a) 50 °C, c) 100 °C, and e) 140 °C for 2 h, respectively. b, d, f) Stress-strain curves of rGO/PPy/PU-UHP after heat treatment at b) 50 °C, d) 100 °C, and f) 140 °C for 2 h, respectively.

In addition, the tensile stress-strain curve of rGO/PPy/PUF1-UHP was measured after heat treatment at 140 °C for 2 h (Supplementary Fig. 49). After heat treatment at 140 °C for 2 h, the elongation at break of rGO/PPy/PUF1-UHP in x direction decreases from 810% to 603%. The decreased elongation at break is probably attributed to the volume expansion after heat treatment.

Supplementary Fig. 49. Tensile stress-strain curve of rGO/PPy/PUF1-UHP in x direction after heat treatment at 140 °C for 2 h.

Based on the above analysis on compressive and tensile stress-strain curves before and after heat treatment at different temperatures, it can be concluded that the mechanical properties of typical hot-pressed porous elastomers remain nearly unchanged after heat treatment at temperatures below 100 °C. Since the thermal insulation and management demonstrations were carried out at the temperature of 51 °C, both rGO/PPy/PUF1-UHP and rGO/PPy/PU-UHP are thermally stable at this applied condition. In addition, the storage moduli of rGO/PPy/PUF1 and rGO/PPy/PU show no obvious decrease in the temperature range of 25-55 °C (Supplementary Fig. 14), indicating that the mechanical properties of the materials are relatively stable at this applied condition.

The above data and discussion have been added to the revised manuscript (Page 18) and supplementary information (Supplementary Figs 47-49 and Supplementary Note 6).

5. As other reviewer mentioned that the group has published similar work, I also checked the work that the authors mentioned in the response letter, I doubt that if the novelty of the present work could meet up with the demand of Nature Communicaitons.

Response:

Thank you for your comments. Our group has published a paper entitled “Bioinspired gradient stretchable aerogels for ultrabroad-range-response pressure-sensitive wearable electronics and high-efficient separators” (10.1002/anie.202213952) (Ref. [22]). However, there are significant differences between this work and our previous work (Ref. [22]), which are shown below.

(1) In this work, we developed a new kind of metamaterials, which were quite different from the aerogels (not metamaterials) in our previous work. Our previous work presented a kind of modulus-gradient aerogels, while this work showed a new kind of porous metamaterials with high stretchability and negative Poisson's ratios.

(2) The preparation methods of these two kinds of materials are significantly different. In our previous work, the gradient aerogel was synthesized by combining three aerogel layers with different moduli. By contrast, the metamaterials in this work were synthesized by novel uniaxial/biaxial/triaxial hot-pressing strategies.

(3) These two kinds of materials show significant difference in their structures. The aerogels in our previous work exhibit a gradient porous structure, while the metamaterials in this work show folded and reentrant microstructures.

(4) The stretchability of the metamaterials (1200%) in this work significantly surpasses those of the aerogels (<100%) in our previous work.

(5) The metamaterials in this work show near-zero and negative Poisson's ratios, which can't be achieved by the aerogels in our previous work.

(6) This work presents broad-range-response strain sensor (1200%) and pressure sensor (1.5 kPa-4.7 MPa), while our previous work only shows a pressure sensor.

(7) We demonstrate that the metamaterials in this work can be used for smart thermal management and electromagnetic interference shielding via stretching, which can't be achieved by the aerogels in our previous work.

As we can see, this work shows high novelty and significant progress over our previous work.

In addition, this work possesses high novelty over other reported stretchable porous materials based on the following reasons.

(1) Various highly compressible aerogels with positive Poisson's ratios can be converted into super-stretchable meta-elastomers with near-zero or negative Poisson's ratios via the versatile uniaxial/biaxial/triaxial hot-pressing strategies, which has never been realized before.

(2) The stretchability of the meta-elastomers significantly surpasses those of the reported stretchable aerogels without specially designed macroscopic structures and is among the highest in those of the reported stretchable porous materials.

(3) The new mechanism of shape fixing during hot pressing and high stretchability of this kind of meta-elastomers are revealed, which will provide important theoretical support for designing highly stretchable metamaterials.

(4) The meta-elastomers can achieve high biaxial (or triaxial) stretchability and negative Poisson's ratios during stretching in different directions, which is difficult to be achieved by traditional negative-Poisson-ratio porous metamaterials.

(5) The maximum detectable tensile strain (1200%) of the meta-elastomer-based strain sensor is among the largest in those of the reported strain sensors based on aerogels, foams, and sponges.

(6) The reversibly tunable thermal insulation and electromagnetic interference (EMI) shielding performances of the meta-elastomers overcome the limitations of the fixed thermal insulation and EMI shielding of traditional porous materials.

The above novelty and significance have been demonstrated in the revised manuscript. The summary of the major novelty and significance of this work has been added to the revised manuscript (Page 19).

Thank you very much for your consideration.

Best regards,

Prof. Guoqing Zu

Department of Polymeric Materials, School of Materials Science and Engineering, Tongji University, Shanghai, P. R. China

REVIEWERS' COMMENTS

Reviewer #2 (Remarks to the Author):

Thanks the author for the kindly responses. My comments are all addressed.